Citation: *Molecular Systems Biology* 9:679
www.molecularsystemsbiology.com

# Insulin/IGF-1-mediated longevity is marked by reduced protein metabolism

Gerdine J Stout[1,4], Edwin CA Stigter[1,4], Paul B Essers[3,4], Klaas W Mulder[2], Annemieke Kolkman[1,5], Dorien S Snijders[1], Niels JF van den Broek[1], Marco C Betist[3], Hendrik C Korswagen[3], Alyson W MacInnes[3] and Arjan B Brenkman[1,*]

[1] University Medical Center Utrecht, Wilhelmina Children's Hospital, Department of Molecular Cancer Research, Section Metabolic Diseases, Utrecht, The Netherlands and Netherlands Metabolomics Centre, Leiden, The Netherlands, [2] Nijmegen Centre for Molecular Life Sciences, Molecular Developmental Biology 274, Nijmegen, The Netherlands and [3] Hubrecht Institute, KNAW and University Medical Center Utrecht, Utrecht, The Netherlands
[4] These authors contributed equally to this work.
[5] Present address: KWR Watercycle Research Institute, Nieuwegein, The Netherlands.
* Corresponding author. University Medical Center Utrecht, Wilhelmina Children's Hospital, Department of Molecular Cancer Research, Section of Metabolic diseases, and Netherlands Metabolomics Centre, Lundlaan 6, Huispostnummer: KC.02.069.1, 3508 AB, Utrecht, The Netherlands; Tel.: + 31 8875 55318; Fax: + 31 8875 54295; E-mail: a.b.brenkman@umcutrecht.nl

Mutations in the *daf-2* gene of the conserved Insulin/Insulin-like Growth Factor (IGF-1) pathway double the lifespan of the nematode *Caenorhabditis elegans*. This phenotype is completely suppressed by deletion of Forkhead transcription factor *daf-16*. To uncover regulatory mechanisms coordinating this extension of life, we employed a quantitative proteomics strategy with *daf-2* mutants in comparison with N2 and *daf-16; daf-2* double mutants. This revealed a remarkable longevity-specific decrease in proteins involved in mRNA processing and transport, the translational machinery, and protein metabolism. Correspondingly, the *daf-2* mutants display lower amounts of mRNA and 20S proteasome activity, despite maintaining total protein levels equal to that observed in wild types. Polyribosome profiling in the *daf-2* and *daf-16;daf-2* double mutants confirmed a *daf-16*-dependent reduction in overall translation, a phenotype reminiscent of Dietary Restriction-mediated longevity, which was independent of germline activity. RNA interference (RNAi)-mediated knockdown of proteins identified by our approach resulted in modified *C. elegans* lifespan confirming the importance of these processes in Insulin/IGF-1-mediated longevity. Together, the results demonstrate a role for the metabolism of proteins in the Insulin/IGF-1-mediated extension of life.
*Molecular Systems Biology* **9**: 679; published online 2 July 2013; doi:10.1038/msb.2013.35
*Subject Categories:* proteomics; proteins
*Keywords:* ageing; high-throughput analysis; metabolism; protein metabolism; translation

## Introduction

The ageing of eukaryotes is subject to environmental and genetic control with conserved features across species, suggesting the presence of common lifespan-regulating mechanisms (Kenyon, 2010). In the nematode *C. elegans*, mutation of the *daf-2* gene encoding for the Insulin/Insulin-like Growth Factor (IGF-1) receptor extends *C. elegans* lifespan two- to three-fold (Kenyon, 2010). The Insulin/IGF-1 pathway is highly conserved and regulates lifespan in organisms ranging from invertebrates to mammals (Kenyon, 2010). Insulin/IGF-1-mediated longevity signalling in *C. elegans* acts exclusively during adulthood, when development has completed (Dillin *et al*, 2002). Interestingly, genetic epistasis analysis has revealed that Insulin/IGF-1-mediated lifespan extension is completely suppressed upon knockdown of a number of transcription factors, including the Forkhead transcription factor DAF-16 (Kenyon *et al*, 1993), the

heat-shock factor HSF-1 (Hsu *et al*, 2003), and partially by the Nrf-like xenobiotic response factor SKN-1(Tullet *et al*, 2008).

Microarray analysis has revealed numerous DAF-16 target genes, including a distinct enrichment of genes involved in stress response (McElwee *et al*, 2003; Murphy *et al*, 2003). These findings are in accordance with the observation that many lifespan-extending mutations concomitantly increase the resistance to stress, including oxidative stress (Wolff and Dillin, 2006). In addition to enhanced stress resistance, Insulin/IGF-1-mediated lifespan extension has been reported to reprogram the ER stress response and to depend on autophagy, the cellular process of self-digestion and recycling (Melendez *et al*, 2003; Hansen *et al*, 2008; Henis-Korenblit *et al*, 2010). Thus, the increased protection of organisms against toxic environmental stress compounded by the activation of autophagy is indispensable to Insulin/IGF-1-mediated longevity.

To date, the biological processes underlying Insulin/IGF-1-mediated longevity remain studied predominantly at the gene level. However, organismal phenotypes are far more dependent on protein function. An initial quantitative proteomics study of Insulin/IGF-1 pathway confirmed the role of stress-protective pathways (Dong *et al*, 2007) during longevity signalling. Additionally, it uncovered several compensatory pathways involved in longevity, underscoring the potential of this approach to identify novel longevity pathways (Dong *et al*, 2007). However, this analysis was restricted to a subset of the nematode proteome, involving mainly cytoplasmic and non-membrane bound proteins (Dong *et al*, 2007). In this study, a more stringent and non-biased proteomics approach of the whole nematode using TMT proteomics was employed (Dayon *et al*, 2008). This recently developed quantification method was used to identify novel processes and pathways involved in Insulin/IGF-1-mediated longevity. The obtained results confirmed the previously reported alteration of several proteins in *daf-2(e1370)* nematodes (Dong *et al*, 2007), including an increased representation of stress-resistance enzymes and a decrease in chaperone proteins. However, our results go on to reveal a severe and previously overlooked reduction in ribosomal proteins and concomitant translational activity. In addition, reduced expression of proteins involved in mRNA processing, translation, and the ubiquitin-proteasome system (UPS) was observed. Functional assays confirmed reduced mRNA levels and 20S proteasomal activity while at the same time total protein content of the mutants compared with wild-type nematodes remained unchanged. Moreover, the importance of these processes for lifespan extension is demonstrated using RNA interference (RNAi)-mediated knockdown of identified candidates.

All together, we propose a model for Insulin/IGF-1-mediated longevity that, in addition to an enhanced stress response, relies on protein metabolism coupled to the reduction in *de novo* protein synthesis and a shift from the UPS of degradation to recycling of proteins via autophagy.

## Results

### Quantitative proteomics of Insulin/IGF-1-mediated longevity

To uncover novel Insulin/IGF-1 lifespan regulators at the protein level, we performed mass spectrometry (MS)-based quantitative proteomics using TMT technology (Dayon *et al*, 2008). TMT involves post-lysis peptide labelling with chemically engineered unique mass tags that appear after peptide fragmentation by MS/MS. These tags allow for accurate protein quantification and when mixed, enable the simultaneous evaluation of protein changes in multiple conditions within a single experiment.

Whole nematode lysates of the long-lived *daf-2(e1370)* Insulin/IGF-1 receptor mutant were compared with wild-type (N2) and *daf-16(mu86); daf-2(e1370)* double mutants (Figure 1A) as the *daf-2*-dependent longevity phenotype is fully suppressed by the *daf-16* mutation (Kenyon *et al*, 1993). As *daf-2(e1370)* is a temperature-sensitive mutant, nematodes were age synchronized at the permissive temperature and allowed to grow to larval stage L4. These nematodes were then switched to the non-permissive temperature (25°C) and harvested at the first day of adulthood when Insulin/IGF-1-dependent lifespan signalling is effective (Dillin *et al*, 2002). Of note, absence of oocytes and eggs was verified by Nomarsky microscopy, to avoid confounding of our observations by a developmental or reproductive phenotype previously reported in the *daf-2(e1370)* mutants (Gems *et al*, 1998).

Total protein extracts from whole nematodes were prepared for each strain in duplicate (6-plex TMT), digested into peptides, and labelled with unique mass tags. Samples were then combined, fractionated to reduce sample complexity, and subjected to LC-MS/MS (Figure 1A). LC-MS/MS analysis resulted in the identification of 599 proteins of which 455 could be quantified (Figure 1B, false discovery rate (FDR) < 5%, Supplementary Figure S1 for representative MS/MS spectrum). Reproducibility of the independent biological replicates for each strain was significant (Supplementary Figure S2, $r \geqslant 0.996$) and proteins were recovered from all major *C. elegans* tissues including nuclear, cytoplasmic, and membrane-bound pools (Supplementary Figure S3).

Major changes were found in the proteome of the *daf-2(e1370)* mutants compared with either N2 ($r = 0.834$) or *daf-16(mu86); daf-2(e1370)* double mutant nematodes ($r = 0.738$, Figure 1B, Supplementary Figure S2). However, little difference was observed between the N2 and *daf-16(mu86); daf-2(e1370)* proteomes ($r = 0.952$, Supplementary Figure S2), in agreement with the observation that *daf-2*-dependent longevity phenotype is completely suppressed by the absence of functional *daf-16* gene expression (Kenyon, 2010).

The abundance of $\sim 40\%$ (cutoff > 1.3 fold change) of all identified proteins was found to be altered in *daf-2(e1370)* mutants (Figure 1B; Supplementary Figure S4). In the *daf-2(e1370)* proteome, 22% of the quantified proteins showed increased abundance, whereas 19% were found decreased as compared with N2. Remarkably, about 90% of all changes depended at least 15–30% on the presence of *daf-16* (Figure 1B), confirming that *daf-16* is indeed a major downstream effector of *daf-2* signalling for the observed proteomic changes.

### Protein translation, proteasomal activity, and mRNA processing are novel pathways downregulated in Insulin/IGF-1 longevity signalling

Over Representation Analysis (ORA) (Backes *et al*, 2007) was used to identify the biological pathways involved in Insulin/IGF-1-mediated longevity. This revealed overrepresentation of stress protective pathways (e.g., thioredoxins and glutathione S-transferases) as well as metabolic pathways (e.g., phosphoenol pyruvate carboxykinases) (Supplementary Figure S5; Supplementary Table I) and confirmed previous studies illustrating the importance of these pathways in Insulin/IGF-1-mediated longevity (Murphy *et al*, 2003). As expected, ORA of proteins with decreased abundance in *daf-2(e1370)* proteome identified pathways with Gene Ontology (GO) classification terms 'ageing' (GO: 0007568) and 'determination of adult lifespan' (GO: 0008340). Growth and

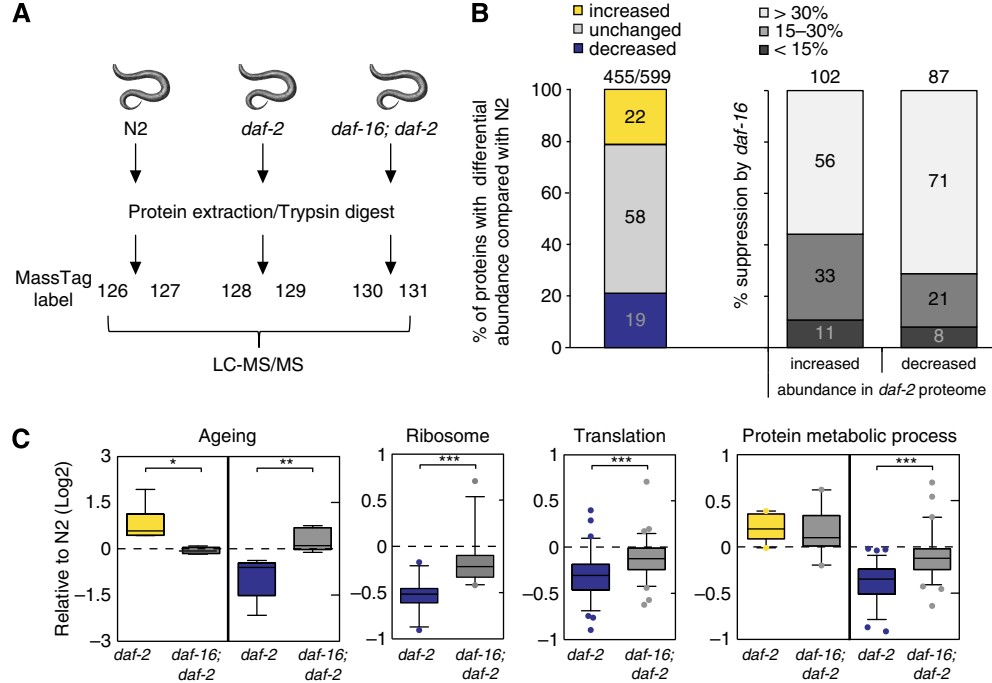

**Figure 1** Quantitative proteomics reveals *daf-16*-mediated reduction in protein metabolism in long-lived *daf-2(e1370)* mutants. (**A**) Schematic overview of experimental set-up. Protein extracts were generated from three sample sets as biological replicates, harvested at day 1 of adulthood. Proteins were trypsin digested and labelled with unique mass tags. Labelled peptides were combined, fractionated by strong cation exchange chromatography to reduce sample complexity and subsequently analysed by LC-MS/MS. Data from 66 fractions were merged and analysed using Proteome Discoverer software. (**B**) Percentage of proteins with differential abundance ($>1.3$ fold change) in *daf-2(e1370)* mutants compared with N2 (left bar), and their expression dependency on *daf-16* (right bars) for proteins with increased and decreased abundance to N2, respectively. The number of quantified proteins out of total amount of identified proteins is indicated on top of the bars. (**C**) Boxplots show expression analysis for GO-enriched processes in *daf-2(e1370)* and *daf-16(mu86); daf-2(e1370)* mutants relative to N2 (Log2). Asterisks indicate statistical significance (\*$P<0.01$, \*\*$P<0.001$, \*\*\*$P<0.0001$) as determined by paired *t*-test analysis.

developmental processes were also found among the most significantly overrepresented GO categories (Figure 1C; Supplementary Figure S5; Supplementary Table II). Analysis using a cutoff-independent statistical method, Gene Set Enrichment Analysis (GSEA), confirmed these observations (Supplementary Table III). Our results are consistent with previous reports (Kenyon, 2010) and despite considerable technological differences, significant concordance was observed between our data set and published microarray (Murphy *et al*, 2003) and proteomics data sets (Dong *et al*, 2007) performed on the Insulin/IGF-1 signalling pathway (Supplementary Figure S6).

Remarkably, significantly enriched and overrepresented GO terms (GSEA and ORA, respectively) that had not previously been associated with Insulin/IGF-1-mediated longevity were identified. A dramatic decrease was observed in proteins belonging to GO terms 'structural constituent of ribosome' (GO: 0003735) and 'ribosome' (GO: 0005840) (Figure 1C; Supplementary Tables II and III). In addition to this reduction in numerous proteins composing the small and large *C. elegans* ribosomal subunits, several translation initiation and elongation factors with decreased abundance were identified (Table I). These observations suggest that the protein translation machinery is significantly reduced in the *daf-2* proteome. Further analysis of proteins with decreased abundance identified the GO terms 'translation' (GO: 0006412), 'protein metabolism' (GO: 0019538) and

'proteasome core complex' (GO: 0005839) as significantly overrepresented in a *daf-16*-dependent manner (Figure 1C; Table I; Supplementary Tables I and III), pointing towards a general reduction in *de novo* protein production and protein turnover in the long-lived *daf-2(e1370)* mutant.

An additional independent quantitative proteomics experiment was conducted, comparing N2 with *daf-2(e1370)* (duplex). The results showed considerable overlap in the results and confirmed our observations (Supplementary Figure S7). Finally, western blot analysis confirmed the quantitative proteomics results for several identified proteins (Supplementary Figure S8). Thus, these findings suggest that Insulin/IGF-1-mediated longevity is associated with altered protein homeostasis.

## Active mRNA translation is diminished in Insulin/IGF-1 mutants

Depletion of specific ribosomal and translation initiation factors has been shown to extend longevity in a range of organisms, including *C. elegans* (Hansen *et al*, 2007; Kapahi *et al*, 2010; McCormick *et al*, 2011). However, reduced protein translation is thought to facilitate Dietary Restriction (DR, defined as reduction in food intake without malnutrition) mediated lifespan extension (Stanfel *et al*, 2009; Kapahi *et al*, 2010), but not Insulin/IGF-1-dependent longevity (Hansen *et al*, 2007).

**Table I** Identified proteins related to protein metabolism that, compared with N2, display reduced abundance in the *daf-2(e1370)* proteome, and are suppressed by *daf-16*.

| Accession | Gene name | NCBI/KOG description | Fold change 6-plex | % DAF-16 suppression |
|---|---|---|---|---|
| *RNA processing* | | | | |
| C44E4.4 | C44E4.4 | RNA-binding protein La | −1.92 | 67.8 |
| Y39A1C.3 | *cey-4* | Predicted RNA-binding protein involved in translation or RNA processing | −1.47 | 41.8 |
| C07H6.5 | *cgh-1* | ATP-dependent RNA helicase (germline/P granule) | −1.64 | 71.0 |
| T01C3.7 | *fib-1* | ScNop1p (U3 SnoRN3P)—Fibrillating and related nucleolar RNA-binding proteins | −1.34 | 28.2 |
| Y65B4BR.5a | Y65B4BR.5 | Transcription factor containing NAC and TS-N domains | −1.49 | 56.7 |
| ZK381.4 | *pgl-1* | Predicted RNA-binding protein that contains a number of C-terminal RGG box motifs | −1.28 | 13.0 |
| *mRNA processing* | | | | |
| Y106G6H.2b | *pab-1* | Polyadenylate-binding protein (RRM superfamily) | −1.49 | 45.8 |
| M28.5 | *phi-9* | 60S ribosomal protein 15.5kD/SNU13, involved in splicing | −1.41 | 45.0 |
| R09B3.3 | R09B3.3 | mRNA cleavage and polyadenylation factor I complex, subunit RNA15 | −2.39 | 101.9 |
| W08E3.1 | *snr-2* | U1 snRNP component | −1.53 | 49.5 |
| F56D12.5a | *vig-1* | Predicted RNA-binding protein (RISC complex—regulation of translation) | −1.48 | 47.0 |
| *Translation* | | | | |
| F28H1.3 | *aars-2* | Alanyl-tRNA synthetase | −1.43 | 48.0 |
| F54H12.6 | *eef-1B.1* | Elongation factor 1 beta/delta chain | −1.33 | 33.7 |
| Y41E3.10a | *eef-1B.2* | Elongation factor 1 beta/delta chain | −1.33 | 33.8 |
| F17C11.9b | *eef-1G* | Translation elongation factor EF-1 gamma/Glutathione S-transferase | −1.36 | 39.0 |
| T23D8.4 | *eif-3.C* | Translation initiation factor 3, subunit c (eIF-3c) | −1.57 | 2.0 |
| C47B2.5 | *eif-6* | Translation initiation factor 6 (eIF-6) | −2.01 | n.a. |
| F57B9.6 | *inf-1* | Translation initiation factor 4a | −1.21 | n.a. |
| *Ribosome* | | | | |
| K07C5.4 | K07C5.4 | Ribosome biogenesis protein—Nop56p/Sik1p | −1.43 | 28.9 |
| C53D5.6 | *imb-3* | Karyopherin (importin) beta 3/nuclear import of ribosomes | −1.30 | 27.6 |
| M28.5 | *phi-9* | 60S ribosomal protein 15.5 kD/SNU13 | −1.41 | 88.0 |
| F13B10.2c | *rpl-3* | 60S ribosomal protein L3 and related proteins | −1.33 | 16.2 |
| B0041.4 | *rpl-4* | 60S ribosomal protein RPL1/RPL2/RL4L4 | −1.53 | 20.8 |
| Y48G8AL.8a | *rpl-17* | 60S ribosomal protein L22 | −1.70 | 30.8 |
| C09D4.5 | *rpl-19* | 60S ribosomal protein L19 | −1.45 | 12.5 |
| E04A4.8 | *rpl-20* | 60S ribosomal protein L18A | −1.44 | 4.7 |
| C27A2.2a | *rpl-22* | 60S ribosomal protein L22 | −1.44 | 9.3 |
| C03D6.8 | *rpl-24.2* | 60S ribosomal protein L30 isologue | −1.87 | 39.9 |
| F52B5.6 | *rpl-25.2* | 60s ribosomal protein L23 | −1.52 | 45.8 |
| F28C6.7a | *rpl-26* | 60S ribosomal protein L26 | −1.37 | 38.7 |
| R11D1.8 | *rpl-28* | 60S ribosomal protein L28 | −1.53 | 36.9 |
| W09C5.6a | *rpl-31* | 60S ribosomal protein L31 | −1.42 | 15.5 |
| C42C1.14 | *rpl-34* | 60s ribosomal protein L34 | −1.35 | 13.8 |
| ZK652.4 | *rpl-35* | 60S ribosomal protein L35 | −1.52 | 25.5 |
| F37C12.4 | *rpl-36* | 60S ribosomal protein L36 | −1.40 | 18.3 |
| C54C6.1 | *rpl-37* | 60S ribosomal protein L37 | −1.38 | 17.5 |
| Y48B6A.2 | *rpl-43* | 60S ribosomal protein L37 | −1.68 | 51.9 |
| C37A2.7 | *rpl-P2* | 60S ribosomal protein | −1.37 | n.a. |
| Y71A12B.1 | *rps-6* | 40S ribosomal protein S6 | −1.43 | 24.3 |
| F42C5.8 | *rps-8* | 40S ribosomal protein S8 | −1.43 | 19.2 |
| T07A9.11 | *rps-24* | 40S ribosomal subunit S24 | −1.36 | 28.3 |
| F39B2.6 | *rps-26* | 40S ribosomal protein S26 | −1.48 | 15.6 |
| *Protein folding* | | | | |
| C47E8.5 | *daf-21* | Molecular chaperone (HSP90 family) | −1.31 | 35.1 |
| F26D10.3 | *hsp-1* | Molecular chaperones HSP70/HSC70, HSP70 superfamily | −1.36 | 31.2 |
| C37H5.8 | *hsp-6* | Molecular chaperones mortalin/PBP74/GRP75, HSP70 superfamily | −1.51 | 38.5 |
| *Protein breakdown* | | | | |
| Y67D8C.5 | *eel-1* | E3 ubiquitin-protein ligase/putative upstream regulatory element binding protein | −1.34 | 26.4 |
| Y110A7A.14 | *pas-3* | 20S proteasome, regulatory subunit alpha type PSMA4/PRE9 | −2.01 | n.a. |
| C36B1.4 | *pas-4* | 20S proteasome, regulatory subunit alpha type PSMA7/PRE6 | −1.32 | 26.2 |
| F25H2.9 | *pas-5* | 20S proteasome, regulatory subunit alpha type PSMA5/PRE | −1.15 | 32.5 |
| CD4.6 | *pas-6* | 20S proteasome, regulatory subunit alpha type PSMA1/PRE5 | −1.81 | 56.1 |
| K05C4.1 | *pbs-5* | 20S proteasome, regulatory subunit beta type PSMB5/PSMB8/PRE2 | −1.46 | 38.4 |
| *Other* | | | | |
| C06G3.5b | C06G3.5 | Adenine deaminase/adenosine deaminase | −3.17 | 313.9 |
| C24F3.2 | C24F3.2 | Dual specificity phosphatase | −6.71 | n.a. |
| ZK863.6 | *dpy-30* | Histone H3 (Lys4) methyltransferase complex, subunit CPS25/DPY-30 | −1.35 | n.a. |
| D2096.8 | *nap-1* | Nucleosome assembly protein NAP-1 | −1.52 | 44.3 |
| C50B6.2 | *nasp-2* | Cell cycle-regulated histone H1-binding protein | −3.50 | n.a. |
| K04D7.1 | *rack-1* | G protein beta subunit-like protein | −1.37 | n.a. |
| ZK742.1a | *xpo-1* | Nuclear transport receptor CRM1/MSN5 (importin beta superfamily) | −2.01 | n.a. |

Values indicate Fold Change (FC) observed in 6-plex TMT experiment, as well as corresponding percentage of *daf-16*-mediated suppression. This list is extended with observations from an independent duplex experiment. n.a., not addressed.

Both longevity pathways genetically interact with signalling by the Target of Rapamycin (TOR) protein complex, which activates several cellular processes, including protein translation (Stanfel *et al*, 2009). Although loss of *let-363*, the *C. elegans* mTOR orthologue, extends lifespan and genetically interacts with *daf-2* (Vellai *et al*, 2003), it was proposed to mediate *C. elegans* Insulin/IGF-1 longevity through upregulation of autophagy rather than reducing protein translation (Melendez *et al*, 2003; Hansen *et al*, 2007, 2008). This, together with the results from ORA and GSEA, prompted us to directly measure protein translation activity in Insulin/IGF-1 mutants.

Actively translated mRNA molecules contain multiple ribosomes (polysomes), whereas translationally inactive mRNA is retained in messenger ribonucleoprotein (mRNP) particles. Polysomes can be quantified after generation of an absorbance profile for rRNA, separated by sucrose gradient velocity sedimentation (Pereboom *et al*, 2011). A profound decrease in the 60S peaks, corresponding to the large ribosomal subunit, as well as a dramatic reduction in polysomal peaks was observed in *daf-2(e1370)* mutants compared with wild-type nematodes (Figure 2A). These findings are consistent with decreased levels of several ribosomal proteins and suggest that the *daf-2(e1370)* mutation drives a decrease in active polysomal protein translation. This decrease is almost completely suppressed by the *daf-16* mutation, illustrating the specific control the Insulin/IGF-1

pathway is able to wield on polysomal mRNA translation (Figure 2A and B) during lifespan extension.

## Insulin/IGF-1 pathway mutants exhibit altered protein metabolism

In addition to a reduced abundance of proteins directly involved in translation, several mRNA processing factors and elements associated with translation initiation were found decreased in the *daf-2(e1370)* proteome (Table I). PAB-1 tightly binds poly(A) tails of mRNA (Sonenberg and Hinnebusch, 2009) and is a key component of the translation initiation complex (Squirrell *et al*, 2006). Interestingly, PAB-1 is present in the cellular mRNA storage bodies (Noble *et al*, 2008) and involved in mRNA transport as well as release of mRNAs from sites of transcription (Dunn *et al*, 2005). Moreover, CGH-1, a putative DEAD box RNA helicase, and CEY-4 (Parker and Sheth, 2007; Updike and Strome, 2010) displayed a *daf-16*-dependent reduction in abundance in the *daf-2(e1370)* proteome (Table I). A similar observation was made for Sm-protein SNR-2, the *C. elegans* U1 subunit orthologue of the spliceosome. Based on these observations, we hypothesized that the global decrease in protein translation is accompanied by a reduction in total mRNA levels. Indeed, using equal numbers of nematodes, the *daf-2(e1370)* mutant yielded significantly less mRNA compared with N2, and this phenotype

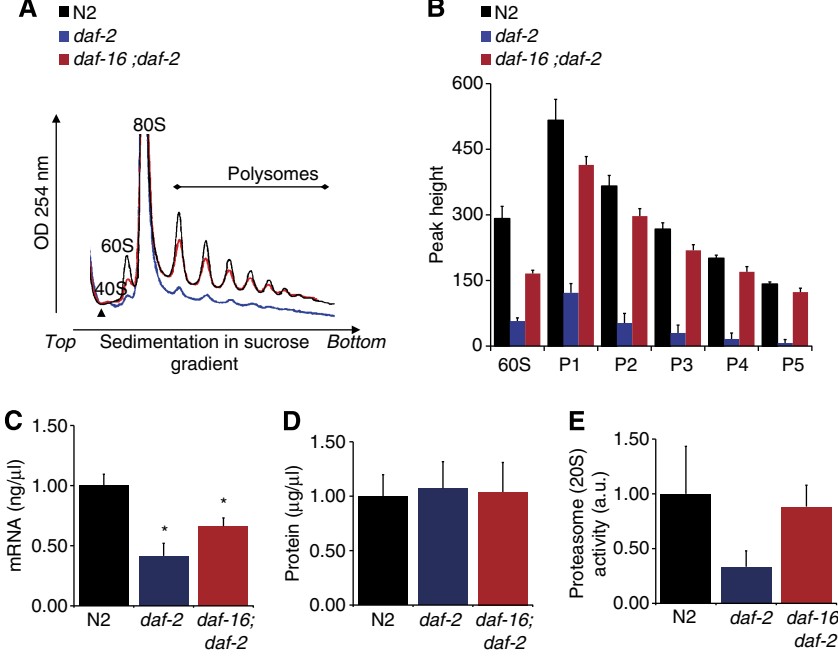

**Figure 2** Functional assays confirm *daf-2*-dependent reduction in protein metabolism. (**A**) Representative traces of polyribosome profiles obtained from N2 (black), *daf-2(e1370)* (blue), and *daf-16(mu86); daf-2(e1370)* (red) nematodes harvested at day 1 of adulthood (identical conditions as described for the proteomics experiments). (**B**) Quantification of the peak (P) heights in the obtained polyribosome profiles using the starting point before the 40S peak (marked by an arrowhead, Figure 2A) as the zero value in at least four independent experiments. Values represent mean ± s.e.m. All measured peaks are significantly lower in *daf-2(e1370)* compared with N2 ($P < 0.0002$) and this response is dependent on *daf-16* ($P < 0.00003$) as tested by Student's *t*-test. a.u., arbitrary units. (**C**) The *daf-2(e1370)* mutant contained significantly less mRNA than N2. mRNA yield (ng/μl) was determined from 1500 nematodes of each strain in triplicate, using poly(A)$^+$ trapping. (**D**) Total protein levels are similar in all three strains. Protein extracts from 1500 synchronized young adults were determined by spectrophotometric analysis. (**E**) The *daf-2(e1370)* has lower 20S proteasomal core complex activity. 20S proteasomal activity was measured in triplicate on 1500 synchronized day 1 adults for each strain using fluorescence analysis. Statistical significant differences represent means ± s.e.m. as calculated with Student's *t*-test (*$P < 0.05$).

was partially suppressed in the *daf-16(mu86); daf-2(e1370)* double mutant (Figure 2C; Supplementary Figure S11). Although the *daf-2 (e1370)* mutant has a 5% smaller body size (McCulloch and Gems, 2003) compared with N2 nematodes at the chosen time point for our experiment, it is unlikely that this small difference in body size can account for the 48% reduction in mRNA levels observed in the *daf-2(e1370)*.

Interestingly, despite reduced mRNA levels and polysomal translation, total protein levels per nematode were unchanged in *daf-2(e1370)* mutants (Figure 2D). We reasoned that reduced protein translation evokes less protein misfolding, ER stress, and reduced protease activity coupled to the constitutive recycling of intrinsic proteins by autophagy. In line with this hypothesis, ORA of proteins with decreased abundance in the *daf-2(e1370)* proteome revealed a significant overrepresentation of proteins involved in protein degradation ('proteasome core complex' (GO: 0005839), Table I; Supplementary Table II). Furthermore, decreased abundance of proteins involved in protein folding (e.g., chaperone DAF-21) and breakdown (E3-ligase, EEL-1 as well as 20S proteasomal subunits (PAS-3, 4, 6, 7, and PBS-5) was observed. This suggests that the *daf-2(e1370)* mutant may display reduced protein degradation activity, pointing to a global protein metabolic phenotype during Insulin/IGF-1-mediated longevity. To investigate this hypothesis, the activity of a main protein degradation pathway, the 20S proteasome (Finley, 2009) was determined. Indeed, reduced activity of this complex was observed in *daf-2(e1370)* mutants, and this response seemed to depend on *daf-16* (Figure 2E). These observations link Insulin/IGF-1-mediated longevity to reduction of proteasomal activity.

Altogether, biological validation of our proteomics results suggests that Insulin/IGF-1-mediated longevity signalling is strongly associated with a global alteration of protein metabolism, in particular downregulation of mRNA processing, protein translation, and protein breakdown activity.

## Reduced protein metabolism regulates longevity

To determine whether the identified processes have an important role in longevity assurance, lifespan assays with RNAi on the identified candidates involved in protein metabolism were performed. siRNA-mediated targeting of *snr-2* and *pab-1* significantly extended median lifespan of wild-type nematodes with 10–30%, indeed suggesting that mRNA transport and processing contribute to *C. elegans* longevity (Supplementary Table IV). It has previously been reported that RNAi-mediated knockdown of several translation initiation and elongation factors, as well as multiple components of the small and large ribosomal subunit, extend *C. elegans* lifespan (Henderson *et al*, 2006; Hansen *et al*, 2007; Reis-Rodrigues *et al*, 2012). Interestingly, many of these proteins showed a decreased abundance in the *daf-2(e1370)* proteome (Table I). Lifespan was also significantly extended with 10–20% when *rpl-17* and *rpl-28* expression is abrogated (Supplementary Table IV).

Knockdown of two identified proteasome subunits significantly decreased lifespan compared with control (Supplementary Table IV) with 10 and 30% for *pas-3* and *pbs-5*, respectively, suggesting that reduced proteasomal activity may induce

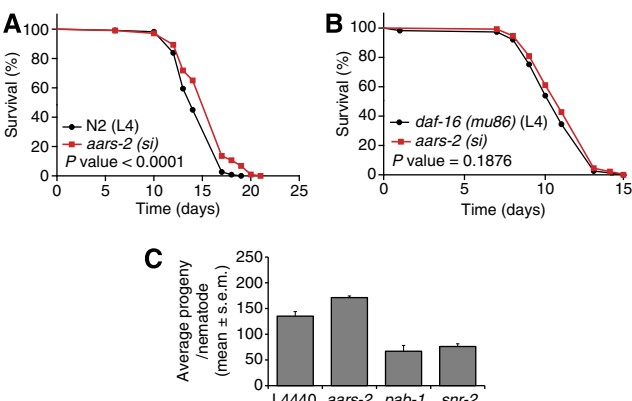

**Figure 3** AARS-2 knockdown extends longevity independent of reduced reproduction, but dependent on *daf-16*. Representative survival curve of N2 (**A**) and *daf-16(mu86)* (**B**) nematodes fed from L4 with either control RNAi (black lines) or RNAi targeting *aars-2*, alanyl-tRNA synthetase (red). All experiments were carried out at 25°C with $n \geqslant 30$ animals per assay. Statistical differences were calculated by log-rank (Mantel–Cox) test. See Supplementary Tables IV and V for the overview of all performed lifespan assays. (**C**) Brood size experiments of N2 nematodes fed on control RNAi or RNAi targeting *aars-2*, *pab-1*, or *snr-2*. Bar graphs represent average offspring per nematode and $N \geqslant 14$ per condition. See Supplementary Table VI for more detail.

toxicity. These findings are in agreement to a previous report that showed decreased lifespan of nematodes with knockdown of proteosomal subunits (Ghazi *et al*, 2007). Finally, abrogation of *aars-2*, a predicted class II aminoacyl-tRNA synthetase that catalyses the attachment of alanine to its cognate tRNA, significantly extended longevity of N2 nematodes by ~10% (Figure 3A; Supplementary Table IV). Together, these findings indicate that downregulation of many proteins involved in mRNA processing, protein translation, and protein breakdown significantly modify *C. elegans* lifespan.

Next, we determined whether the extended longevity was mediated by transcription factor *daf-16/FOXO*. Indeed, knockdown of all tested RNAi constructs failed to extend lifespan of the *daf-16(mu86)* mutants (Supplementary Table V). It was also discovered that the long lifespan of *daf-2(e1370)* could not be further extended by knockdown of *rpl-17* and *rpl-28*. These observations suggest that the *daf-16*-dependent longevity pathway activity in *daf-2(e1370)* mutants overlaps with the longevity signalling induced by knockdown of the *rpl-17* and *rpl-28* ribosomal proteins.

The strong *daf-2(e1370)* mutant has previously been described to show reduced fecundity at 25°C, independently from its effects on longevity (Gems *et al*, 1998). Nematodes for the proteomics experiments were harvested as young adults, before presence of progeny (oocytes and eggs) production. Nevertheless, we found a number of proteins significantly downregulated in the *daf-2(e1370)* mutant for which the expression is either localized or enriched to the germline, including PGL-1, CGH-1, PAB-1, CAR-1, and CEY-4 (Table I). Several of these proteins were found to mediate lifespan upon RNAi knockdown (Supplementary Table IV). These findings suggest that at least part of the longevity phenotype of the *daf-2(e1370)* is mediated by reduced germline activity. We therefore performed brood size experiments to determine whether the reduced protein metabolism phenotype is an epi-

phenomenon of the reduced fecundity. The reduced fecundity phenotype of the *daf-2(e1370)* mutant was partially dependent on *daf-16* as shown by the significant recovery of brood size in the *daf-16(mu86); daf-2(e1370)* double mutant (Supplementary Figure S9; Supplementary Table VI). Knockdown of both SNR-2 and PAB-1 expression significantly reduced fecundity in the N2 (Figure 3C), in agreement with the strong expression of both proteins in the germline. In contrast, fecundity was unaffected upon siRNA-mediated knockdown of *aars-2* which is predominantly expressed in the nematode intestine and hypodermis. As RNAi-mediated knockdown of *aars-2* extends longevity (Supplementary Table IV), this result suggests that longevity and reproduction can be uncoupled at the level of protein metabolism regulation.

Next, polyribosome analysis was performed in the *daf-2(e1370)* mutant propagated at 15 and 20°C when a longevity phenotype is observed either in the absence (15°C) or with a small (20°C) reproductive effect (Gems *et al*, 1998); (Kenyon *et al*, 1993). A reduction in active protein translation was found (Supplementary Figure S10), although the magnitude of the reduction tracks with the severity of the reproductive phenotype.

These observations do not fully exclude the possibility that the germline is involved in mediating the observed *daf-16*-dependent protein metabolism phenotype. To investigate this possibility, polyribosome profiling experiments were performed in germline-ablated nematodes. To this end, *daf-2(e1370)* and N2 nematodes were fed siRNA against GLP-1, a germline-specific GLP-/Notch receptor (Arantes-Oliveira *et al*, 2002). Interestingly, polyribosome profiles of N2 nematodes did not markedly differ upon abrogation of GLP-1 (Figure 4A), a condition that has been reported to induce lifespan extension (Arantes-Oliveira *et al*, 2002). Despite the fact that global protein translation of the *daf-2(e1370)* mutant was strongly reduced compared with N2 nematodes (Figures 2A and 4), polyribosome profiles of germline ablated and control *daf-2(e1370)* mutants did not markedly differ (Figure 4B). Together, these findings indicate that the observed global changes in protein translation in Insulin/IGF-1 signalling are unlikely to result from reduced reproduction or germline activity.

In summary, these observations provide evidence of a causal role for altered protein metabolism in Insulin/IGF-1-mediated longevity, and additionally suggest that protein metabolism, rather than protein translation alone, governs Insulin/IGF-1-mediated longevity assurance.

## Discussion

### Insulin/IGF-1-mediated longevity is associated with altered protein metabolism

The performed unbiased quantitative proteomics study of Insulin/IGF-1 signalling pathway mutants revealed that Insulin/IGF-1-mediated longevity is strongly associated with a global reduction in protein metabolism at the levels of mRNA processing, protein translation, and protein breakdown (Figures 1 and 2; Table I; Supplementary Figure S5; Supplementary Tables I–III). Moreover, reduced expression of multiple proteins (through RNAi) involved in these

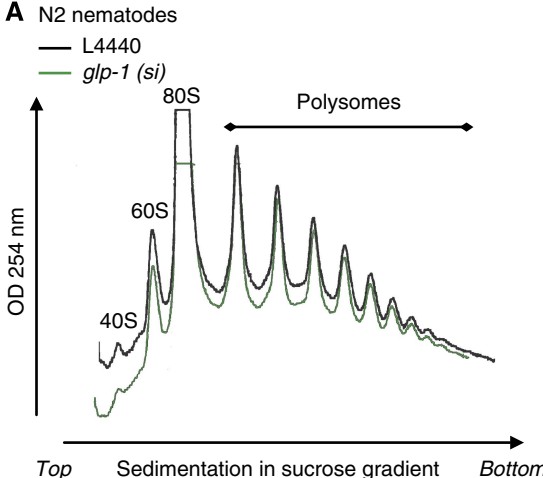

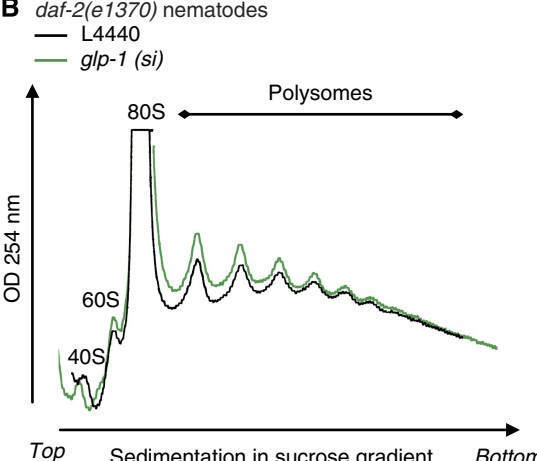

**Figure 4** Germline ablation does not affect active protein translation. Representative traces of polyribosome profiles obtained from (**A**) N2 nematodes and (**B**) *daf-2(e1370)* mutants fed on control (black line) or *glp-1* siRNA (green line). Nematodes were synchronized, cultured on *glp-1* siRNA from L1 stage, switched to 25°C from L4 and harvested at day 1 of adulthood. Observed minor differences in profiles reflect experimental variation.

processes extended *C. elegans* lifespan in a *daf-16*-dependent manner (Figure 3; Supplementary Tables IV and V), indicating that these proteins contribute to Insulin/IGF-1-mediated longevity.

An initial proteomics study on the *C. elegans* Insulin/IGF-1 pathway revealed comparable results to our study (Supplementary Figure S6), however, failed to identify the massive reduction in ribosomal proteins reported here (Dong *et al*, 2007). The experiment described by Dong *et al* was restricted to a subset of the nematode proteome, involving mainly cytoplasmic and non-membrane bound proteins (S-100 fraction), thereby likely missing the ribosomal protein fraction bound to the ER or localized in the nucleolus.

A role for protein translation in Insulin/IGF-1-mediated lifespan regulation has not been demonstrated to date. This is underscored by the fact that *de novo* incorporation of labelled methionine in *daf-2(e1370)* and wild-type nematodes did not differ, thus arguing against a role for decreased protein

translation in Insulin/IGF-1-mediated longevity (Hansen *et al*, 2007). However, as recognized by Hansen *et al*, a possible contribution from protein degradation to their phenotype could not be ruled out due to the absence of a kinetic analysis. As a result, the authors reported a balance between protein synthesis and breakdown rather than active protein translation alone (Hansen *et al*, 2007). The observations from direct assays as reported here (Figure 2), e.g. lower abundance of core 20S proteasome subunits and decreased proteasomal activity, concomitant with strongly decreased protein translation in *daf-2(e1370)* mutants, resolves this apparent paradox, particularly since equal total protein levels were observed in all tested strains. Thus, the *daf-2(e1370)*-mediated longevity phenotype involves both reduced protein translation and proteasomal degradation.

We propose that reduced protein translation in the Insulin/IGF-1 pathway leads to less protein misfolding, protein aggregation, and subsequent ER stress. We envisage that, in turn, the demand for proteasomal breakdown is reduced resulting in high fidelity protein homeostasis. In support of this hypothesis, *daf-2(e1370)* mutants indeed show lower 20S activity (Figure 2E) and moreover, two recent quantitative proteomics analysis of insoluble proteins that gradually accumulate during normal ageing revealed an age-related decline in protein homeostasis (David *et al*, 2010; Reis-Rodrigues *et al*, 2012). It has additionally been established that increased ER stress and protein aggregation strongly reduce longevity, processes which depend on Insulin/IGF-1 signalling (Henis-Korenblit *et al*, 2010; Taylor and Dillin, 2011; Reis-Rodrigues *et al*, 2012). Therefore, we propose that preservation of total protein levels in *daf-2(e1370)* mutants, despite reduction in total mRNA and translational activity, is due to the previously established increase in autophagy activity (Melendez *et al*, 2003; Hansen *et al*, 2008) coupled to our novel results demonstrating abatement of protein degradation.

## Reproduction and Insulin/IGF-1-mediated longevity

The *daf-2(e1370)* proteome also disclosed an altered abundance of several proteins involved in reproduction (Supplementary Figure S5; Table I; Supplementary Tables II and III), including proteins like CGH-1, PGL-1, CAR-1, PAB-1, and SNR-2, which all show expression specific to or enriched in the germline (Squirrell *et al*, 2006; Parker and Sheth, 2007; Buchan and Parker, 2009; Sonenberg and Hinnebusch, 2009; Updike and Strome, 2010). This observation is in line with the delayed reproduction and reduced fertility phenotype displayed by the *daf-2(e1370)* mutant at the non-permissive conditions used in our proteomics study, in addition to its longevity phenotype (Gems *et al*, 1998). Interestingly, RNAi-mediated knockdown of several of these proteins enhanced longevity (Supplementary Tables IV and V). These results therefore suggest that the longevity phenotype of *daf-2(e1370)* mutants at the non-permissive temperature can be at least partially explained by reduced germline function, as germline ablation extends *C. elegans* lifespan (Arantes-Oliveira *et al*, 2002). This notion raises the question as to whether the reduced protein metabolism phenotype could be a mere

epi-phenomenon of the reduced fecundity reported for the *daf-2(e1370)* mutant (Gems *et al*, 1998; Supplementary Figure 9). However, several lines of evidence indicate that the observed longevity effects mediated by reduced protein metabolism activity are distinct from effects on the germline. First, abrogation of ribosomal proteins did not further extend the lifespan of *daf-2(e1370)* mutants (Supplementary Table V), consistent with previous work from the Kenyon laboratory (Hansen *et al*, 2007), whereas germline ablation does extend the lifespan of *daf-2(e1370)* nematodes (Arantes-Oliveira *et al*, 2002). Second, knockdown of the predominantly somatically expressed *aars-2* extended the longevity in the absence of a reproduction phenotype (Figure 3; Supplementary Tables IV and VI). Finally, while germline ablation extends longevity (Arantes-Oliveira *et al*, 2002), this occurs independently of changes in global protein translation (Figure 4).

Altogether, these findings argue that the protein metabolism phenotype identified in Insulin/IGF-1 signalling stems predominantly from somatic tissue. Interestingly, this is consistent with the observation that the germline is not required for lifespan extension of the eukaryotic soma-specific IFE-2 isoform of mRNA translation initiation factor 4E (Syntichaki *et al*, 2007). Of note, germline signalling has been genetically linked to a putative translation elongation factor (Ghazi *et al*, 2009) and it therefore remains possible that germline ablation-mediated longevity signals through specific changes in protein metabolism that occur below the detection limits of our polysome profiling assay.

## Regulation of reduced protein metabolism in Insulin/IGF-1-mediated longevity

The concordance between our results and previous microarray studies (McElwee *et al*, 2003; Murphy *et al*, 2003) suggests good correlation between gene and protein expression of relevant genes (Supplementary Figure S6). Of note, a conspicuous absence of correlation with respect to regulation of protein translation genes was noticed, particularly regarding ribosomal proteins even when specifically queried (McElwee *et al*, 2003). Our data indicated that the reduced abundance of ribosome proteins was at least partially dependent on *daf-16* (Supplementary Tables II and III), but ribosomal protein genes have not been identified as direct DAF-16 targets (Oh *et al*, 2006). These findings suggest longevity to be regulated at the post-transcriptional/translational level rather than exclusive regulation of gene expression. TOR signalling, which was recently identified to relay its effect on longevity through SKN-1/Nrf and DAF-16/FOXO, is an attractive candidate pathway to mediate such regulation given its conserved role in protein translation control and longevity regulation across species, including *C. elegans* (Kapahi *et al*, 2010; McCormick *et al*, 2011; Robida-Stubbs *et al*, 2012). This hypothesis is supported by the observation that lifespan of *daf-2(e1370)* mutants is not further extended by knockdown of the *C. elegans* TOR orthologue, *let-363*, suggesting overlapping signalling pathways (Vellai *et al*, 2003).

TOR mediates protein homeostasis via regulation of mRNA translation initiation, as well as ribosome biogenesis. Interestingly, several proteins involved in translation initiation

(e.g., EIF-3.C, EIF-6, PAB-1, and INF-1), as well as ribosomal biogenesis have been identified as significantly reduced in the *daf-2(e1370)* proteome (Table I; Supplementary Tables II and III). Therefore, reduced TOR signalling could explain the observed reduced protein translation in *daf-2(e1370)* nematodes, although several reported observations are inconsistent with such a model. For instance, in higher organisms TOR and Insulin/IGF-1 signalling are linked through the tuberous sclerosis proteins TSC1/2, thereby regulating processes like protein synthesis for cell growth. However, no orthologues of these proteins have thus far been identified in *C. elegans* and reduced Insulin/IGF-1 signalling in *C. elegans* does not change cell size (Finch and Ruvkun, 2001). Moreover, *let-363* RNAi decreased $^{35}$S-Methionine incorporation in N2 nematodes in line with reduced protein translation, whereas the *daf-2(e1370)* mutant did not (Hansen *et al*, 2007), suggesting that alternative or additional mechanisms are involved. In contrast to TSC1/2, the GTPase RHEB-1 is conserved in *C. elegans* and bridges the GTPase activating protein (GAP) activity of the TSC1/2 complex to TOR activity in multiple organisms (Aspuria and Tamanoi, 2004). Although a role for RHEB-1 in protein translation has not been demonstrated directly in *C. elegans*, RHEB-1 has been shown to mediate longevity induced by intermitted fasting (IF) in a *daf-16*-dependent manner. Interestingly, this supports the concept of molecular coupling between the IF-induced longevity and the Insulin/IGF-like signalling pathway (Honjoh *et al*, 2009).

A translation initiation factor (Syntichaki *et al*, 2007) and several ribosomal proteins have recently been described to extend longevity in a *daf-16*-independent manner (Hansen *et al*, 2007). In contrast, other proteins involved in protein synthesis have recently been shown to mediate longevity through *daf-16* signalling (Henderson *et al*, 2006; Hansen *et al*, 2007; Tohyama *et al*, 2008). The latter is in agreement with our observations revealing that the expression, translation, and function of all tested proteins involved in protein metabolism mediate longevity in a *daf-16*-dependent manner (Figure 1B; Table I; Supplementary Table V). This apparent paradox may be explained by several recent observations, which indicate that *daf-16* signalling is complex. For instance, the extent of DAF-16 knockdown may be important, since most downregulated proteins involved in Insulin/IGF-1 protein metabolism are only partially suppressed by *daf-16* (Table I). Moreover, knockdown of translation elongation initiation factors eIF-4G and eiF2b extends longevity in a *daf-16*-dependent manner when expression is suppressed in adulthood, and independent of *daf-16* when suppressed during development (Henderson *et al*, 2006), suggesting that the timing of *daf-16* suppression is important. Lastly, additional proteins or modifications could be involved in regulating the *daf-16* dependency. Indeed, although *let-363* knockdown was shown to mediate longevity in a *daf-16*-independent manner (Vellai *et al*, 2003), closer inspection of the TOR complexes TORC1 and TORC2 recently revealed that reduced TORC1 activity extends longevity in a *daf-16*-dependent manner (Robida-Stubbs *et al*, 2012). Future biochemical and genetic analysis will be required to characterize how Insulin/IGF-1 signalling is coupled to protein metabolism in *C. elegans*.

## Protein translation as a common longevity assurance mechanism

Insulin/IGF-1- and DR-mediated longevity have long been considered as distinct processes (Lakowski and Hekimi, 1998; Houthoofd *et al*, 2003; Min *et al*, 2008), however, others have recently suggested that Insulin/IGF-1- and DR-mediated longevity may share mechanistic features (Clancy *et al*, 2002; Greer *et al*, 2007; Narasimhan *et al*, 2009; Kenyon, 2010). The work presented here is in strong support of the latter model, and moreover extends this parallel between Insulin/IGF-1- and DR-mediated longevity since DR-mediated longevity was recently shown to also depend on decreased protein translation in *D. melanogaster* (Zid *et al*, 2009) and *S. cerevisiae* (Kapahi *et al*, 2010; McCormick *et al*, 2011). In fact, reduced expression of proteins involved in translation has been shown to extend longevity in a range of animals, including *S. cerevisiae* (both chronological and replicative models), *D. melanogaster*, and *C. elegans* (McCormick *et al*, 2011). Moreover, knockdown of the mTOR-responsive S6K and rapamycin administration (both known to inhibit protein translation) extended longevity in mice (Harrison *et al*, 2009; Selman *et al*, 2009). Also as previously mentioned, longevity mediated by germline signalling was genetically linked to a putative translation elongation factor (Ghazi *et al*, 2009). Together with our findings, we propose that protein homeostasis provides a common denominator in longevity assurance. In support of this hypothesis, a genome-wide linkage analysis performed on a cohort of human individuals with exceptional longevity identified a locus on chromosome 4 that exerts substantial influence on the ability to achieve exceptional old age (Puca *et al*, 2001). Interestingly, the human orthologue of *RPL34* directly flanks this locus. Our proteomics study reveals that abundance of rpL34 is decreased in *daf-2(e1370)* mutants (Table I; Supplementary Table II), while abrogation of *rpl-34* was found to contribute to *C. elegans* longevity (Hansen *et al*, 2007). It is therefore attractive to speculate that human longevity assurance may also be orchestrated through the fundamental regulation of protein translation.

## Concluding remarks

This study demonstrates that Insulin/IGF-1-mediated longevity assurance is directly linked to decreased protein translation, which draws a parallel to DR-mediated longevity and suggests that these longevity mechanisms share mechanistic features. Additionally, reduced mRNA processing and protein degradation is shown to contribute to an overall decreased protein metabolism phenotype and Insulin/IGF-1-mediated longevity. These findings support a role for decreased protein translation in Insulin/IGF-1-mediated longevity and potentially as a common longevity assurance mechanism.

## Materials and methods

### Nematode growing and conditions

#### Strains
All strains were maintained expanded as described previously (Brenner, 1974). N2, *daf-2(e1370)*, *daf-16(mu86)*, and *daf-16(mu86)*:

*daf-2(e1370)* were obtained from the *Caenorhabditis* Genetics Center (CGC). Nematodes were propagated on 150 mm NGM OP50 plates at 15–20°C.

## RNAi clone analysis

All RNAi containing bacteria (HT115) clones, except for the *daf-2* and *cgh-1* siRNA which were kind gifts of Dr D Gems (University College, London, UK) and Dr TK Blackwell (Harvard Medical School, Boston, USA) respectively, were purified from Ahringer's or Vidal libraries (Kamath *et al*, 2003; Rual *et al*, 2004). The identity of all RNAi clones was verified by sequencing. Clones were grown overnight at 37°C in LB containing 100 µg/ml ampicillin before seeding.

## Germline ablation

Synchronized L1 nematodes were plated on vector control or *glp-1* RNAi-containing HT115 bacteria plates, incubated at 15°C until the worms reached L4 stage, then temperature shifted to 25°C for 16–20 h, harvested and subsequently snap frozen for polysome profiling. Ablation of the germline of adult nematodes was confirmed by Nomarsky microscopy before the nematodes were harvested.

## Lifespan analysis

Lifespan analysis was performed as described previously (Hansen *et al*, 2005) and conducted at 25°C. In brief, strains were synchronized by hypochlorite treatment and isolated eggs were grown on 150 mm NGM OP50 plates at 20°C until late L3/early L4 stage. Of note, the development of obtained synchronized N2, *daf-2(e1370)*, and *daf-16(mu86)* eggs/larvae was delayed by placing them 16–20 h at 15°C to compensate for the slight delayed development of the *daf-2(e1370)* nematodes (Hirsh *et al*, 1976; Gems *et al*, 1998). In all, 10–12 nematodes in L4 or young adult (YA) stage were picked and passed onto 10–12 35 mm NGM plates inoculated with the gene-specific RNAi bacteria (HT115) of interest, pre-cultured as described above. After picking, the nematodes were switched to 25°C for lifespan evaluation. The pre-fertile period of adulthood was chosen as $T = 0$ for lifespan analysis. Strains were grown at optimal growth conditions (20°C) for at least two generations before lifespan analysis was started. The NGM plates used for lifespan were supplemented with 100 µg/ml ampicillin, 40 µM IPTG, and 200 µg/ml 2′fluoro-5′deoxyuridine (FUDR) (all obtained from Sigma, St Louis, MO, USA) to induce siRNA expression and to restrict progeny development during the lifespan assay, respectively. Nematodes were counted daily until all nematodes were deceased. An animal was scored dead when it no longer responded to (mechanical) probing/stimulation. Blind scoring was followed for all experiments. Censoring in the lifespan analysis included animals that ruptured, bagged (i.e., exhibited internal progeny hatching), or crawled off the plates. Statistical significance was calculated using the log-rank (Mantel-Cox) method.

## Brood size experiments

Brood size experiments were performed essentially as described previously (Gems *et al*, 1998). Synchronized eggs were incubated at 15°C until worm reach L4 stage, than placed on fresh plates individually and temperature shifted to 25°C. For the RNAi experiments, nematodes in L4 stage were placed NGM RNAi-containing HT115 bacteria plates, which targeted indicated RNAi constructs. The nematodes were transferred to fresh plates twice a day, and the average number of progeny was scored in a blind manner. Statistical significance was calculated using Student's *t*-test with Welch correction.

## Quantitative proteomics

### Peptide generation

Nematodes of all three strains were age synchronized as described. In brief, eggs were collected after hypochlorite treatment and propagated until L4 larval stage at 20°C, except for N2 and *daf-2(e1370); daf-16(mu86)* which were placed for 16–20 h at 15°C to compensate for the reported delayed growth phenotype of *daf-2(e1370)* as described (Hirsh *et al*, 1976; Gems *et al*, 1998). At early L4 larval stage, all three strains were shifted to 25°C for 16–20 h. After visual confirmation of absence of the oocytes, eggs, and progeny by Nomarsky microscopy the nematodes were harvested.

Nematode pellets (about 12 000 day 1 adults per biological replicate) were homogenized and the proteins were extracted in protein extraction buffer (8 M Urea, HALT protease and HALT phosphatase inhibitors (Thermo) with a BulletBlender, using 0.5 mm Zirconium Oxide beads (NextAdvance) for 3 min. After 10 min on ice, extracts were centrifuged and protein concentrations were determined using Nanodrop (Thermo) technology at OD280. Equal amounts of protein (100 µg) were reduced by Tris(2-carboxyethyl)-phosphine hydrochloride for 1 h at 55°C, alkylated by iodoacetamide for 30 min at RT in dark conditions, diluted and digested o/n with trypsin (Thermo). Next, peptides mixtures were labelled with TMT labelling 6-plex kit (Thermo) according to manufacturer's instructions, merged in equal quantities and dried in a vacuum centrifuge.

## Liquid chromatography and MS analysis

Complexity of peptide mixture was reduced by Strong Cation Exchange (SCX) chromatography on an Accela HPLC system (Thermo) coupled to a 2996 Photodiode Array Detector (Waters) using a PolySulfethyl A SCX column (100 mm × 2.1 mm i.d., 5 µm, 300 Å, PolyLC, Columbia, MD) over a 25-min linear gradient from 0 to 40% B at a constant flow rate of 200 µl/min (Buffer A = 5 mM $KH_2PO_4$, 30% acetonitrile and 0.05% formic acid, Buffer B = Buffer A supplemented with 350 mM KCl). A total of 33 fractions were collected off-line every minute. The collected samples were dried in a vacuum centrifuge and dissolved in 20 µl 5% FA. Next, samples were subjected to nano-flow LC (Eksigent) using a $C_{18}$ reverse phase trap column (Phenomenex; column dimensions 2 cm × 100 µm, packed in-house) and subsequently separated on $C_{18}$ analytical columns (Reprosil-Pur C18-AQ 5µm; column dimensions, 20 cm × 50 µm; packed in-house) using a linear gradient from 0 to 40% B (Buffer A = 0.1 M acetic acid; Buffer B = 95% (v/v) acetonitrile with 0.1 M acetic acid) in 60 min and at a constant flow rate of 150 nl/min. Column eluate was analysed directly in a coupled LTQ-Orbitrap-XL mass spectrometer (Thermo Fisher Scientific) operating in positive mode, using internal Lock mass calibration. For optimal identification and quantification, samples were run in both CID and HCD fragmentation mode.

## Data analysis and quantification

All 66 data sets (33 fractions analysed in CID and HCD mode) from 6-plex experiment were merged, processed, and quantified using Proteome Discoverer (Thermo) software and subjected to database searches (Mascot, version 2.3.01, Matrixscience) against a *C. elegans* database (Wormprep 2.12: www.wormbase.org). For the database search, up to two missed trypsin cleavages were allowed with a 10-p.p.m. precursor mass tolerance and 0.8 Da for the fragment ion. Cysteine carbamidomethylation was set as a fixed modification and oxidation of Methionine and TMT labelling on Lysine and $NH_2$ terminus was set as variable modifications. Assembly of individual peptides into proteins follows 5% FDR as peptide confidence threshold and require at least one unique peptide. The vast majority of peptides were identified multiple times in sequential SCX fractions. Relative protein quantification was performed using Proteome Discoverer software (version 1.2.0.208) according to manufacturer's instructions on either the two or six reporter ion intensities per peptide. Redundant peptides, peptides lacking >2 mass tag values or ratios were all excluded.

The MS data used for this publication have been submitted Peptide Atlas (http://www.peptideatlas.org/PASS/PAS00192) and assigned the identifier (PASS00192).

## Bioinformatics analysis

Over/under-representation analysis (ORA) and GSEA were performed using web-based tool GeneTrail (Backes *et al*, 2007). Significance (cutoff $P < 0.05$) was determined by a hypergeometric distribution test using all *C. elegans* genes as background with a 5% FDR multiple

testing correction. Genes with a 1.3-fold change in *daf-2(e1370)* mutants relative to wild type were included in the analysis. The results of the ORA were subsequently confirmed by GSEA, a method that is independent of any fold change cutoff. We refer to Supplementary Tables I–III for full overview of ORA and GSEA results.

## Polyribosome profiling

Gradients of 17–50% sucrose (11 ml) in gradient buffer (110 mM KAc, 20 mM MgAc$_2$ and 10 mM HEPES pH 7.6) were prepared on the day before use in thin-walled, 13.2 ml, polyallomer $14 \times 89$ mm centrifuge tubes (Beckman-Coulter, USA). Nematodes were lysed in 500 µl polysome lysis buffer (gradient buffer containing 100 mM KCl, 10 mM MgCl$_2$, 0.1% NP-40, 2 mM DTT and 40 U/ml RNasin; Promega, Leiden, Netherlands) using a dounce homogenizer. The samples were centrifuged at 1200 *g* for 10 min to remove debris and the supernatant was subjected to protein content analysis by Bradford reagent (Biorad). In all, 610 µg of total protein for each sample was loaded atop the sucrose gradients. The gradients were ultra-centrifuged for 2 h at 40 000 r.p.m. in a SW41Ti rotor (Beckman-Coulter, USA). The gradients were displaced into a UA6 absorbance reader (Teledyne ISCO, USA) using a syringe pump (Brandel, USA) containing 60% sucrose. Absorbance was recorded at an OD of 254 nm. All steps of this assay were performed at 4°C or on ice and all chemicals came from Sigma-Aldrich (St Louis, MO, USA) unless stated otherwise.

## Western blotting

Proteins were extracted from nematode pellets and the protein concentration was measured, as described before. In all, 40–200 µg of protein was separated on SDS–PAGE and transferred onto polyvinylidene difluoride membrane (PVDF) membrane (Immobilon; Millipore, Billerica, MA). Western blot analysis was performed in duplicate under standard conditions (1/500 in 2.5% milk/2.5% BSA) using the indicated antibodies. For quantification of protein levels, membranes were probed with HRP-conjugated secondary antibodies (1/10 000 in 2.5% milk/2.5% BSA). Immunocomplexes were detected using enhanced chemiluminescence (ECL-Plus GE Healthcare) and ImageQuant™ LAS 4000 bimolecular imager with provided software according to the manufacturer (GE Healthcare Europe, Diegem, Belgium).

## Antibodies

Polyclonal antibodies raised against proteasome 20 α-subunits (ab22674), ribosomal protein L22 (sc-98857) and L28 (sc14151) and Lasu1/Ureb1 (A300-486A) were purchased from Abcam (3) (Cambridge, UK), Santa Cruz Biotechnology (2) (Santa Cruz, CA, USA), and Bethyl Laboratories (1) (Montgomery, TX, USA), respectively. Rabbit polyclonal antibody recognizing actin (20–33) was obtained from Sigma-Aldrich.

## mRNA extraction

mRNA was extracted from 1500 young adults for each strain using Dynabeads mRNA DIRECT Mini Kit (610.11, Invitrogen, Oslo, Norway) according to manufacturer's instructions in triplicate. mRNA yield was determined using Nanodrop (Thermo) technology at OD260. Student's *t*-test with Welch correction was used to test the null hypothesis.

## 20S Proteasome activity assay

Nematode pellets (1500 day 1 adults) were homogenized and the proteins were extracted under native conditions in NP-40 extraction buffer (1% Nonidet P-40; 50 mM Tris–HCl, pH 8; 150 mM NaCl; 0.1% SDS; 0.5% SDS). After 10 min on ice, extracts were centrifuged and the protein concentrations were determined using Nanodrop technology at OD280. 20S proteasome activity was determined in triplicate using the 20S proteasome activity Assay Kit according to manufacturer's

instructions (Chemicon International, Inc.). The assay is based on detection of the fluorophore 7-Amino-4-methylcoumarin (AMC) after proteasomal cleavage from the labelled substrate LLVY-AMC. After 2 h of incubation at 37°C, the free AMC fluorescence, a measure for proteasomal activity, was quantified using fluorometer equipped with 380/450 nm filter sets. Statistical significance was calculated with Student's *t*-test with Welch correction.

## Supplementary information

## Acknowledgements

The Gems and Blackwell laboratories are acknowledged for providing *daf-2* and *cgh-1* siRNA bacteria strains, respectively. Members of the Brenkman laboratory are acknowledged for stimulating discussions. GJS, AK, and ECAS were financed by grants from the Netherlands Metabolomics Centre which is part of the Netherlands Genomics Initiative/Netherlands Organisation for Scientific Research.

*Author contributions:* GJS, ECAS, PBE, AK, NFJB, and DSS carried out the experiments. MB and NFJB provided technical assistance. GJS, ECAS, PBE, and KWM performed the analysis of the results. GJS, ECAS, AWM, and ABB wrote and edited the manuscript. AWM, HCK, and ABB supervised the study. GJS, ECAS, RK, and ABB designed the study.

## Conflict of interest

The authors declare that they have no conflict of interest.

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
