## [Review Process File · Molecular Systems Biology]

Insulin/IGF-1 mediated longevity is marked by reduced protein metabolism

Gerdine J. Stout, Edwin C. A. Stigter, Paul B. Essers, Klaas W. Mulder, Annemieke Kolkman, Dorien S. Snijders, Niels J.F. van den Broek, Marco C. Betist, Hendrik C. Korswagen, Alyson W. MacInnes, Arjan B. Brenkman

Corresponding author: Arjan Brenkman, UMC Utrecht

Review timeline:

Submission date:	08 August 2012
Editorial Decision:	26 September 2012
Revision received:	25 January 2013
Editorial Decision:	11 March 2013
Revision received:	24 April 2013
Editorial Decision:	21 May 2013
Revision received:	24 May 2013
Accepted:	27 May 2013

Editors: Andrew Hufton/Thomas Lemberger

Transaction Report:

1st Editorial Decision

26 September 2012

Thank you again for submitting your work to Molecular Systems Biology. We have now heard back from the two referees who agreed to evaluate your manuscript. As you will see from the reports below, the referees find the topic of your study of potential interest. They raise, however, substantial concerns on your work, which, I am afraid to say, preclude its publication in its present form.

The referees both agreed that this work could be valuable to the field, but they had fundamental concerns which they felt currently cast serious doubts on the main conclusions of this work. The editor would like to emphasize, that given the importance of these concerns, substantial additional experimental work appears to be needed to convincingly address these issues. The first issue, raised by both reviewers, was a concern that changes in germ line tissue, fecundity, or developmental dynamics could potentially explain the observed changes in translation and proteasome related protein levels. Addressing this issue appears to require a thorough evaluation of the tissue specificity of the observed proteomic changes, and to rigorously rule out developmental contributions. Secondly, both reviewers raised more general technical concerns regarding experimental replication and the need for appropriate controls. Once again, addressing this issue appears to require additional experimental work, and may require repeating some of the main experiments with additional controls.

Please make sure that the Supplementary Figures and Tables are supplied with the revised work as a single pdf file separate from the main manuscript. Supplementary Tables with more than 50 rows should be supplied as separate files in either Excel or tab-delimited text formats. In addition, we generally require authors to deposit all new proteomic datasets in a public repository, and to make these data available to reviewers during the review process. We therefore ask that you to deposit

these data in a suitable repository, such PRIDE or PeptideAtlas, and supply a confidential reviewer login with any revised work.

If you feel you can satisfactorily deal with these points and those listed by the referees, you may wish to submit a revised version of your manuscript. Please attach a covering letter giving details of the way in which you have handled each of the points raised by the referees. A revised manuscript will be once again subject to review and you probably understand that we can give you no guarantee at this stage that the eventual outcome will be favorable.

 REFEREE REPORTS:

Reviewer #1 (Remarks to the Author):

This is an informative and for the most part nicely presented study that describes an extensive quantitative proteomic analysis of the insulin/IGF-1 pathway. The authors report that daf-2 mutant *C. elegans* are characterized by reduced levels of many proteins that are associated with mRNA translation, among other processes not detected in a previous study. Notably, these differences are dependent upon daf-16, which is essential for the effects of daf-2 inhibition on longevity and other processes. They also show evidence that these animals have reduced protein translation and proteasome activity, while maintaining similar total protein levels to wild-type animals. Taken at face value, the paper provides a very exciting and provocative new model to help explain how the daf-2 pathway influences aging. However, in addition to some smaller concerns, two very important issues must be addressed before the work can be published.

The first and most fundamental problem is that the authors must present evidence and/or airtight arguments that the effects they show on translation-associated proteins and processes are indeed happening in somatic tissues, where they would need to be in play to be involved in daf-2 effects on aging. The study examined adult *C. elegans*, in which approximately half of the "cells" are found in the germline, in which most of them are syncytial. It is well-known that the daf-2(e1370) mutant affects reproduction independently of its effects on longevity, and that translation regulation and presence of mRNA binding proteins are conserved and central aspects of oogenesis. Seen in this light, it is worrisome that among the proteins detected as being lower in these mutants are PGL-1 and CGH-1, which respectively are expressed specifically or almost completely in the germline. Similarly, in many species PAB-1 and Y-box proteins (CEY-4) are expressed at particularly high levels in developing germ cells. If daf-2 is reducing levels of these (as almost certainly predicted) and ribosomal/translation proteins (certainly possible) largely in germ cells, this would be an epiphenomenon that would not be involved in its longevity effects. It is also worrisome that the Ruvkun lab has published that daf-2 mutants express pgl-1 and other germline genes at abnormally high levels in somatic cells. Was this not detected here because the reductions occurred predominantly in the germline? It seems that it would be necessary at least to show that translation and expression of some relevant proteins are reduced in somatic cells in the daf-2 context.

Also, the lifespan experiments must be strengthened considerably. It needs to be shown that the effects are reproducible across multiple independent experiments. This is particularly crucial when the effects are very modest, as is the case here. In each case, it must be documented that the appropriate control (N2) was analyzed simultaneously in each experiment. Many of the results shown are not convincing at all, and even those that are stronger seem to derive from only a single experiment each. Also, it would have been nice if more new genes had been analyzed, particularly from the proteasome since this result would be surprising if demonstrated convincingly.

Additional concerns:

- The authors state that although it was previously published that daf-2 animals do not have reduced de novo incorporation of labeled methionine, this is in-line with their finding that daf-2 animals have reduced polysome formation and ribosomal protein, given their finding that daf-2 animals have reduced proteasome activity - but these do not seem to add up. Decreased active polysome translation and concurrent decreased proteasome activity could explain why they found daf-2 animals have similar protein levels to those of wild-type animals, however wouldn't the authors' findings still suggest that there would be a decrease in labeled methionine incorporation, or

translation, in *daf-2* animals during a particular window in time? This needs to be resolved or explained better.

- Regarding figure three, the authors can conclude that knock-down of their identified targets increases lifespan (though by how much, should be specified in the text), however on page 13, they cannot claim that these factors mediate IIS longevity, or title the last section on page 12, "Protein metabolism governs Insulin/IGF-1 mediated longevity," or title the paper similarly. The effects observed were incredibly modest, especially when compared with those of *daf-2*.
- Citation of the literature on translation, TOR, and lifespan in *C. elegans* is somewhat superficial. For example, it is not mentioned that the role of *daf-16* has been controversial with respect to translation, or that TOR RNAi extends lifespan independently of *daf-16*, which would have implications for the models being proposed here.
- The term "deregulation," used several times in the paper, is a bit misleading. The regulation of protein homeostasis in *daf-2* animals, is not lacking, it is altered.
- On page seven, the authors conclude the middle paragraph by stating the 90% of all proteome changes in the *daf-2* animals depended "to some extent" on the presence of *daf-16*. While the term "to some extent" is vague, "90%" is not. It would therefore be more accurate to say: about 90% of all changes depended at least 15-30% on the presence of *daf-16*.
- On page eleven, at the end of the first paragraph, it should be mentioned whether any size differences between N2, *daf-2*, and *daf-16;daf-2* animals could account for the different levels of mRNA.
- The discussion is very long and speculative, and does not adequately discuss the findings themselves.

Reviewer #2 (Remarks to the Author):

The MS reports a study of changes in expression of proteins in insulin pathway mutants in *C. elegans*. The aim was to discover how reduced insulin signaling, which extends the lifespan of the worm, alters expression of proteins, and which of these changes are dependent upon *daf-16*, which is required for lifespan extension by reduced insulin signaling in *C. elegans*. This is not the first study to look at changes in protein expression in this context, but the use of a more advanced methodology and MS platform has allowed a deeper coverage of the proteome. The current results are similar to previous ones, and largely accord with those of RNA expression profiles of these mutants, although there is an increased emphasis on the role of protein synthesis and, particularly, turnover.

The main problem with the study is that, as with previous studies of gene expression in the worm, there is a major complication from changes in tissue composition in the mutants. These mutants affect the timing of development and reproduction and the fecundity of the adult. Much of the signal that is derived from changes in gene expression in the whole organism may therefore reflect changes in maturation and tissue composition, rather than altered function of specific tissues. The same comment applied to the assay of proteasome activity - it is not clear where the signal is coming from.

SPECIFIC COMMENTS:

This manuscript investigates the proteome of the long-lived *daf-2* mutant. Using *daf-2;daf-16* double mutants, the investigators addressed the issue of which components of the proteome determine lifespan in a *daf-16*-dependent manner. The relatively new technique of Tandem Mass Tagging (TMT) was used to quantify differences in the proteome, and the data indicated a down-regulation of translation and proteasomal activity. To validate these results, the authors examined lifespan in worms with reduced (via RNAi) components of mRNA processing, translation, and protein breakdown. Such an approach could lead insights into IIS-mediated lifespan-extension, but there are some technical issues with the work.

1. Both proteome and polysome analysis of a whole organism can be greatly affected by changes in tissue composition. *daf-2* mutants have approximately 10% (Gems et al 1998) to 75% (Voorhies and Ward 1999) reduced lifetime fecundity. Total fecundity and egg laying rate in *daf-2* mutants is extremely sensitive to temperature changes (Gems et al 1998) with *daf-2* mutants spreading egg production over a longer egg laying period than that of wild type. If total fecundity and egg laying rate is reduced, at any specific time point, overall translation is also likely reduced and this would be strongly reflected in the proteome and polysome profile (as it seems to be in figure 2A).

Furthermore, *daf-2* mutants are developmentally delayed and mature more slowly. Day 1 adults were used for the analysis and, even with a suppressor of egg-development present, there will certainly have been differences in maturation and reproductive state between genotypes. This could explain a lot of the difference in protein synthesis and turnover.

2. The concordance between replicates for the proteome analysis was high, but what about the polysomes? Did the polysome profile traces show significantly replicable patterns? One picture is not data.
3. RNA levels (fig 2c) were directly compared between equal numbers of worms but this could again be affected by developmental stage and fecundity.
4. Figure 3, were the lifespan experiments single experiments, if so the n value is very low, and the experiments should be replicated with more worms, and scored blind.
5. How was the proteome analysis normalized? Could taking 610ug of total protein from each sample bias the analysis? N2 and *daf-2* mutants are different sizes (lengths and volume).
6. Was a parallel life span experiment run along side those worms sampled for proteome/profiling/proteasome experiments?
7. It is reasoned that ER stress and autophagy may be altered in the *daf-2* mutants. Why were these not measured?
8. What temperature were worms grown at for figure 2? 20 or 25 degrees? Or switched as with the lifespan?
9. Figure 2c-e SEM should also be calculated for all N2 samples to show sampling variation.
10. Supp. Figure 3b Gonad, not Gonade
11. Supp. Fig 8 - were the westerns repeated? Although these data are confirmatory, the blots should be quantified and analyzed appropriately.
12. Lifespans should also be carried out in a *daf-2*, *daf-2*;*daf-16* null background to show dependence/or not.
13. Figure 3-table 3a, the level of knockdown should be shown - qPCR or western.
14. Supp. Figure 6 - does 'other' mean unchanged in both sets? Or is there only a 5% overlap with the Murphy study- not in high concordance as suggested?
15. Does the addition of FUDR or Ampicillin affect lifespan? Either by regulating DNA replication (endoreduplication or egg production?) or gut fauna? Is there an effect on lifespan of using non-dividing OP50?

RESPONSE TO REVIEWERS

Manuscript number : MSB-12-3940-T

Title: Protein metabolism governs Insulin/IGF-1 mediated longevity in *C. elegans*

Detailed response to REVIEWER #1:

The reviewer concludes that: "Taken at face value, the paper provides a very exciting and provocative new model to help explain how the daf-2 pathway influences aging. However, in addition to some smaller concerns, two very important issues must be addressed before the work can be published.

We greatly appreciate the reviewers' enthusiasm and in the detailed overview below we have addressed all his/her concerns point-by-point, and additionally indicated where the manuscript has been adapted accordingly:

Reviewer #1 - 1: The first and most fundamental problem is that the authors must present evidence and/or airtight arguments that the effects they show on translation-associated proteins and processes are indeed happening in somatic tissues, where they would need to be in play to be involved in daf-2 effects on aging. The study examined adult *C. elegans*, in which approximately half of the "cells" are found in the germline, in which most of them are syncytial. It is well-known that the daf-2(e1370) mutant affects reproduction independently of its effects on longevity, and that translation regulation and presence of mRNA binding proteins are conserved and central aspects of oogenesis. Seen in this light, it is worrisome that among the proteins detected as being lower in these mutants are PGL-1 and CGH-1, which respectively are expressed specifically or almost completely in the germline. Similarly, in many species PAB-1 and Y-box proteins (CEY-4) are expressed at particularly high levels in developing

germ cells. If daf-2 is reducing levels of these (as almost certainly predicted) and ribosomal/translation proteins (certainly possible) largely in germ cells, this would be an epi-phenomenon that would not be involved in its longevity effects.

It is also worrisome that the Ruvkun lab has published that daf-2 mutants express pgl-1 and other germline genes at abnormally high levels in somatic cells. Was this not detected here because the reductions occurred predominantly in the germline? It seems that it would be necessary at least to show that translation and expression of some relevant proteins are reduced in somatic cells in the daf-2 context.

Answer: *We thank the reviewer for calling our attention to this important point and here provide the requested airtight arguments, as well as experimental evidence, illustrating that the presented observations are not simply an epi-phenomenon of the reduced translation in the germ line:*

- *Germ cell inhibition extends the already long lifespan of daf-2(e1370) (Arantes-Oliveira et al, 2002; Hsin & Kenyon, 1999). To rule out that inhibition of translation extends longevity simply via germ cell inhibition, we performed RNAi mediated targeting of rpl-17 and rpl-28 in the daf-2(e1370) mutant and found that longevity was however not extended, whereas rpl-17 and rpl-28 knockdown extended lifespan of N2 nematodes (Suppl. Table V and VI). These findings are consistent with previous work from Hansen et al (Hansen et al, 2007), who show that RNAi of none of their tested translation factors, including ribosomal proteins, extended daf-2(e1370) lifespan. These observations argue against the possibility that lifespan extension is mediated by germ cell inhibition.*
- *One of the identified candidates with reduced abundance in daf-2(e1370) proteome, an alanyl-tRNA synthase (aars-2), is required for protein biosynthesis. This protein is ubiquitously expressed in the nematode, but predominantly in the intestine and hypodermis. RNAi mediated knockdown of aars-2 expression extends lifespan of N2 nematodes (Fig. 3, Suppl. Table IV). Fecundity experiments that were performed on the tested mutant strains, as well as upon RNAi mediated knockdown of several of our candidates revealed that reduced fecundity is not affected upon*

knockdown of *aars-2* expression (**Fig. 3, S9, Suppl. Table VI**). This suggests that *aars-2* mediates its effects on lifespan via somatic tissue. In addition, the absence of a fecundity phenotype concomitant with an extended lifespan upon knockdown of *aars-2* additionally suggests that reduced fecundity alone cannot explain the extended lifespan observed in the *daf-2(e1370)* mutant.

- Next to the data reported for 25°C, polyribosome analysis (PRA) on *daf-2(e1370)* mutants was performed on nematodes propagated at permissive temperatures 15°C and 20°C ($n \geq 3$). At these temperatures fecundity is wild type-like while longevity phenotype is retained (Gems et al, 1998). PRA revealed that active translation is reduced and a drop in polyribosomes peaks is observed (**Suppl. Fig. S10**). These findings suggest that a reduction in active translation is associated with the observed longevity independently of the germ line.

Results from this additional set of experiments have been incorporated into the manuscript (page, 12), as well as Fig. 3, Fig. S9-S11 and Suppl. Table VI) and the discussion section has been adapted accordingly (page 16).

Reviewer #1 - 2: Also, the lifespan experiments must be strengthened considerably. It needs to be shown that the effects are reproducible across multiple independent experiments. This is particularly crucial when the effects are very modest, as is the case here. In each case, it must be documented that the appropriate control (N2) was analyzed simultaneously in each experiment. Many of the results shown are not convincing at all, and even those that are stronger seem to derive from only a single experiment each.

Also, it would have been nice if more new genes had been analyzed, particularly from the proteasome since this result would be surprising if demonstrated convincingly.

Answer: Lifespan assays for most of the identified candidates have been repeated with increased nematode numbers per experiment and experiments were scored blindly. The appropriate control has been included with each round of experiments and obtained results are now included in **Suppl. Table IV**, where we have specifically indicated the corresponding controls (numbers; column 4) required for comparison. The same holds for lifespan experiments performed in *daf-16(mu86)* mutant backgrounds (see point reviewer #2 - 12).

Indeed, for many candidates, the lifespan extension upon RNAi mediated knockdown is modest in comparison to the *daf-2(e1370)* mutant. However, it is anticipated that the cumulative lifespan effect of many of the individually RNAi targeted proteins related to protein metabolism, as well as other processes affected in the *daf-2(e1370)* (**Fig. 1c**), contribute, at least in part, to the long lifespan of the *daf-2(e1370)* mutant.

As suggested by the reviewer we have tested several more candidates, including proteasome and RNA body candidates. Several performed lifespan analysis using RNAi for *pbs-5* and *pas-3* point to a previously unappreciated decrease in longevity consistent with previous work from the Kenyon lab (Ghazi et al, 2007), suggesting that depletion of the proteasome components induces a toxic effect on *C. elegans* lifespan and, therefore, we cannot conclusively attribute a role to proteasomal degradation in longevity. Finally, lifespan assays that were performed with novel candidates, *cgh-1* and *pgl-1*, revealed a significant lifespan extension with RNAi mediated knockdown.

The results from additional work performed for the revision of the manuscript have been incorporated into the manuscript in results (page 11) and discussion section (page 18), as well as Suppl. Table IV, V.

Reviewer #1 - 3: The authors state that although it was previously published that *daf-2* animals do not have reduced de novo incorporation of labeled methionine, this is in-line with their finding that *daf-2* animals have reduced polysome formation and ribosomal protein, given their finding that *daf-2* animals have reduced proteasome activity - but these do not seem to add up. Decreased active polysome translation and concurrent decreased proteasome activity could explain why they found *daf-2* animals have similar

protein levels to those of wild-type animals, however wouldn't the authors' findings still suggest that there would be a decrease in labeled methionine incorporation, or translation, in daf-2 animals during a particular window in time? This needs to be resolved or explained better.

Answer: *The conclusion of the ³⁵S-labeling experiment performed by Hansen et al. is discussed in the presented manuscript since a possible role for reduced protein translation in Insulin/IGF-1 mediated longevity has been dismissed based on their observations (Hansen et al, 2007), in turn generating an apparent paradox to our observations. We consider the conclusion presented by Hansen et al. preliminary as the performed ³⁵S-labeling experiment was carried out at a single time point only, thus lacking information on kinetics of the translation process. Moreover, Hansen et al. acknowledge that applied ³⁵S-labeling assay reports the balance between protein synthesis and breakdown. Thus, rather than an apparent paradox, this balanced synthesis/breakdown explanation of the single time-point (Hansen et al, 2007) is consistent with our direct observations of both active translation repression concomitant with decreased 20S proteasome activity in the daf-2(e1370) mutant. In our revised manuscript we have now rephrased the third paragraph of the discussion (page 14-15) to further explain this reasoning.*

Reviewer #1 - 4: Regarding figure 3, the authors can conclude that knock-down of their identified targets increases lifespan (though by how much, should be specified in the text), however on page 13, they cannot claim that these factors mediate IIS longevity, or title the last section on page 12, "Protein metabolism governs Insulin/IGF-1 mediated longevity," or title the paper similarly. The effects observed were incredibly modest, especially when compared with those of daf-2.

Answer: *Here we would like to add that candidate proteins were identified based on the reduced abundance in daf-2(e1370) mutant. Lifespan assays were indeed performed in N2 background to verify an anticipated effect on the lifespan, we therefore conclude that the observed lifespan effect is associated to the Insulin/IGF1 pathway. We have now specified the observed significant, however modest, lifespan effects in the text (page 11), and re-titled both the paragraph and the manuscript (page 11 and page 1, respectively).*

Reviewer #1 - 5: Citation of the literature on translation, TOR, and lifespan in *C. elegans* is somewhat superficial. For example, it is not mentioned that the role of daf-16 has been controversial with respect to translation, or that TOR RNAi extends lifespan independently of daf-16, which would have implications for the models being proposed here.

Answer: *We thank the reviewer for calling our attention to this gap in our manuscript and have adapted the revised manuscript to emphasize the role of TOR signaling (page 17-18) as well as the signaling of daf-16 with respect to protein translation and longevity (page 18).*

Reviewer #1 - 6: The term "deregulation," used several times in the paper, is a bit misleading. The regulation of protein homeostasis in daf-2 animals, is not lacking, it is altered.

Answer: *We followed the reviewer's suggestion and the word 'deregulation' has been replaced by 'altered' in throughout revised manuscript.*

Reviewer #1 - 7: On page seven, the authors conclude the middle paragraph by stating the 90% of all proteome changes in the daf-2 animals depended "to some extent" on the presence of daf-16. While the term "to some extent" is vague, "90%" is not. It would therefore be more accurate to say: about 90% of all changes depended at least 15-30% on the presence of daf-16.

Answer: *We followed the reviewer's suggestion and now state on page 6, end of 2nd paragraph: "about 90% of all changes depended at least 15-30% on the presence of daf-16".*

Reviewer #1 - 8: On page eleven, at the end of the first paragraph, it should be mentioned whether any size differences between N2, daf-2, and daf-16; daf-2 animals could account for the different levels of mRNA.

Answer: *Indeed, the daf-2(e1370) mutant has a slightly longer body length (6.3%) but slightly shorter body width (5.9%) compared to the N2, resulting in about 5% smaller body size (McCulloch & Gems, 2003). This relative small difference in body volume is unlikely to account for the ~50% reduction of mRNA levels observed in the daf-2(e1370) as shown in **Fig. 2c**.*

We have followed the reviewers suggesting and included a remark regarding this concern in the manuscript on pages 9-10).

Reviewer #1 - 9: The discussion is very long and speculative, and does not adequately discuss the findings themselves.

Answer: *We have taken the reviewers concern into consideration and have rewritten part of the discussion. Through focusing on our observations (quantitative proteomics data as well as direct assays), including those from the additional work performed for the revision of the manuscript, the speculative character of the discussion was reduced. The discussion of the revised manuscript has however been extended with recently published literature with respect to the role of TOR signaling (pages 16-18) (see above #1 - 5)*

Detailed response to REVIEWER #2:

Reviewer 2 appreciates the insights our work provides towards understanding IIS-mediated lifespan extension, however raises some technical issues. In the detailed overview below we have addressed all specific comments point-by-point, and indicated where we have adapted the manuscript accordingly. In particular:

Reviewer #2 - 1: The main problem with the study is that, as with previous studies of gene expression in the worm, there is a major complication from changes in tissue composition in the mutants. These mutants affect the timing of development and reproduction and the fecundity of the adult. Much of the signal that is derived from changes in gene expression in the whole organism may therefore reflect changes in maturation and tissue composition, rather than altered function of specific tissues. The same comment applied to the assay of proteasome activity - it is not clear where the signal is coming from. *And in addition:* Both proteome and polysome analysis of a whole organism can be greatly affected by changes in tissue composition. *daf-2* mutants have approximately 10% (Gems et al 1998) to 75% (Voorhies and Ward 1999) reduced lifetime fecundity. Total fecundity and egg laying rate in *daf-2* mutants is extremely sensitive to temperature changes (Gems et al 1998) with *daf-2* mutants spreading egg production over a longer egg laying period than that of wild type. If total fecundity and egg laying rate is reduced, at any specific time point, overall translation is also likely reduced and this would be strongly reflected in the proteome and polysome profile (as it seems to be in figure 2A).

Furthermore, *daf-2* mutants are developmentally delayed and mature more slowly. Day 1 adults were used for the analysis and, even with a suppressor of egg-development present, there will certainly have been differences in maturation and reproductive state between genotypes. This could explain a lot of the difference in protein synthesis and turnover.

Answer: *The reviewer argues that the observed phenotype of reduced protein metabolism must be uncoupled from the other phenotypes i.e. delayed maturation and reproduction. Regarding development it is important to emphasize that our experiments were performed on age-synchronized nematodes. To compensate for the slower development rate of the *daf-2(e1370)* nematodes, we followed a commonly applied strategy resulting in temporary delay in the development of the wild-type and the *daf-16; daf-2* double mutants. To this end we placed synchronized egg preps (achieved through hypochlorite treatment) of the mutants at 15°C for several hours (16-20 hours), while *daf-2(e1370)* mutants were propagated at 20°C from seeding. All strains were then propagated at 20°C until shift to the permissive temperature (25°C) for *daf-2(e1370)* mutation. Such propagation scheme is set up based on the published growth rate tables and development tables (Gems et al, 1998; Hirsh et al, 1976). Of note, the nematodes cannot be grown at 25°C from seeding due to a constitutive dauer phenotype present in *daf-2(e1370)* mutant.*

*Moreover, we have selected young adults, which are largely post-developmental, for all our analysis, except lifespan assays, and in all cases assured that the worms were synchronized populations (applied the above described temporal temperature shift for N2 and *daf-16;daf-2* nematodes) and did not have any visible signs of eggs and progeny.*

We have adapted the materials and methods section (page 21) and now explain this more clearly in the results section (page 5, paragraph 2).

*With respect to the concerns raised regarding the reproductive phenotype observed in the *daf-2(e1370)* mutant we would like to refer to the answer provided to the comment of reviewer #1 - 1. The observations from the performed work carried out to answer this question have been incorporated into the manuscript in results (page 12) and discussion section (page 16), as well as Fig. 3, Fig. S9, Fig. S10 and Table VI.*

Reviewer #2 - 2: The concordance between replicates for the proteome analysis was high, but what about the polysomes? Did the polysome profile traces show significantly replicable patterns? One picture is not data.

Answer: *The polyribosome profiles indeed show very high concordance and reproducibility. Although we only present 1 representative profile trace in Fig. 2a, the accompanying graph (Fig. 2b) is the result of 4 independently performed experiments with a total of 4 independently collected pellets per strain as mentioned in the legend of Fig. 2b (page 36).*

Reviewer #2 - 3: RNA levels (fig 2c) were directly compared between equal numbers of worms but this could again be affected by developmental stage and fecundity.

Answer: *Here we would like to refer the answer provided regarding point 1 and of both reviewers #1 and #2 (page 1 and 5 of this document, respectively)*

Reviewer #2 - 4: Figure 3, were the lifespan experiments single experiments, if so the n value is very low, and the experiments should be replicated with more worms, and scored blind.

Answer: *We followed the reviewer's suggestion and increased both the number of experiments as well as the number of nematodes per experiment. All experiments have been carried out using blind scoring. We additionally refer to the answer provided to reviewer #1 – 2, page 2.*

The obtained results are now included in a new Fig. 3 and Suppl. Table IV of the revised manuscript which now reflects the new data (page 11), and the materials and methods section has been completed (page 21-22).

Reviewer #2 - 5: How was the proteome analysis normalized? Could taking 610ug of total protein from each sample bias the analysis? N2 and daf-2 mutants are different sizes (lengths and volume).

Answer: *The proteome analysis was normalized for total protein concentration as described in the methods section "peptide generation", using equal amounts of protein (100 µg) for each strain as input material. It was noted that 60% of all proteins (including structural and housekeeping proteins) remain unchanged between the daf-2(e1370) and the N2 (Fig. 1b) and that many identified proteins with altered abundance are known regulators of Insulin/IGF-1 mediated lifespan as determined by other groups and based on independent experimental approaches. Of note, polyribosome analyses were also normalized to total protein concentration to allow comparison, and equal amounts of proteins (610ug) were loaded onto sucrose gradients.*

Regarding the possible bias due to nematode size differences we would like to refer to point 1 of reviewer #2). In brief, the daf-2 (e1370) mutant has an approximate 5% smaller body volume (McCulloch & Gems, 2003) compared to the N2 at the selected time-point for harvesting (young adult without visible signs of eggs/progeny). This relative small size difference is unlikely to account for the prominent reduction of mRNA levels (~50% reduction) and >30% reduced protein expression of proteins specifically involved in protein metabolism that was observed in the daf-2(e1370) mutants (Fig. 2, Table I, Suppl. Table I-III).

Moreover, the three tested strains contained equal protein levels when isolated from a fixed number of nematodes, suggesting that the protein concentration per nematode is not different for the studied strains (Fig. 2), which included the daf-16(mu86); daf-2(e1370) double mutant that represses nearly all phenotypes related to a mutation in daf-2 (Dillin et al, 2002; Gems et al, 1998).

In the revised manuscript we have now included a remark regarding body size differences (page 9-10) and the unlikeliness thereof to account for the observed effects.

Reviewer #2 - 6: Was a parallel life span experiment run along side those worms sampled for proteome/profiling/proteasome experiments?

Answer: *The lifespan phenotypes of the daf-2(e1370), N2 and daf-16(mu86); daf-2(e1370) strains are very reproducible and robust (see Suppl. Table IV and V). All proteome/profiling/proteasome experiments were performed in triplicate at least, and for each experiment at least one lifespan assay was run in parallel.*

Reviewer #2 - 7: It is reasoned that ER stress and autophagy may be altered in the daf-2 mutants. Why were these not measured?

Answer: *Both altered autophagy and ER stress are established phenotypes in the daf-2 mutants. Here we would like to refer to literature published on this subject (Hansen et al, 2008; Henis-Korenblit et al, 2010; Melendez et al, 2003); references of our revised manuscript. However, we realized that these topics deserved to be introduced better. Therefore, we have added the ER stress response, next to the autophagy section, in the introduction of the manuscript (page 3, paragraph 2).*

Reviewer #2- 8: What temperature were worms grown at for figure 2? 20 or 25 degrees? Or switched as with the lifespan?

Answer: *The worms for this experiment were indeed switched to 25°C at L4 stage, as described for the lifespan. This has been emphasized in the in the legend of Fig. 2a (page 36).*

Reviewer #2 - 9: Figure 2c-e SEM should also be calculated for all N2 samples to show sampling variation.

Answer: *We follow the reviewers suggestion and have corrected this in the revised figure 2c-e.*

Reviewer #2 - 10: Supp. Fig 3b Gonad, not Gonade

Answer: *This has been corrected.*

Reviewer #2 - 11: Supp. Fig 8 - were the westerns repeated? Although these data are confirmatory, the blots should be quantified and analyzed appropriately.

Answer: *The Western analyses serve as additional confirmation of the two independent quantitative proteomics studies, involving three independent biological replicates. The Western experiments were performed in biological duplicates of independently harvested nematode pellets. A representative experiment, including quantification, is shown in Suppl. Fig. S8.*

In the revised legend of Suppl. Fig. 8 we have further clarified the quantification of this data and the material and methods section now includes information on the amount of repetitions (page 26).

Reviewer #2 - 12: Lifespans should also be carried out in a daf-2, daf-2;daf-16 null background to show dependence/or not.

Answer: *The proposed experiment unfortunately is technically difficult to perform particularly since the daf-2 null mutant has a nonconditional dauer phenotype at 25°C prevents its use in lifespan assays (Gems et al, 1998).*

We did however conduct lifespan assays on the daf-2(e1370) mutant in combination with RNAi for the identified ribosomal proteins rpl-17 and rpl-28. Whereas knockdown of both proteins modestly increased N2 lifespan (Suppl. Table IV), they did not further increase the long lifespan of daf-2(e1370) (2 independent experiments per RNAi). These observations suggest that the longevity pathways of the daf-2(e1370) and rpl-17 and rpl-28 knockdown are overlapping, in agreement with the reports by Hansen et al (Hansen et al, 2007). Thus, abrogation of rpl-17 and rpl-28 mediates at least partially daf-2(e1370) longevity. In the revised manuscript this new data has been added (Suppl. Table V) and the results and discussion section has been adapted according on pages 12 and 16, respectively.

In order to test the dependency of the observed lifespan effects on daf-16 lifespan assays in daf-16(mu86) mutants were conducted, also upon siRNA mediated knockdown (2 independent experiments per RNAi) of our identified candidates. All tested RNAi constructs that increased the lifespan on N2 nematodes (Suppl. Table IV), did not extend lifespan on the daf-16(mu86) mutant confirming a dependency on daf-16 (Suppl. Table V). These findings support the presented proteomics data which reveal that all identified candidate proteins found with a reduced abundance in the daf-2(e1370) mutants, and which extended lifespan upon knockdown by RNAi in the wild-type, were suppressed at least partially by daf-16 (Table 1, Suppl. Table II, IV, V). The revised manuscript is adapted to reflect these new results accordingly on pages (11-12 and 16, 18) and Suppl. Table V has been added to the revised Supplementary Information Files.

Reviewer #2 - 13: Figure 3-Table 3a, the level of knockdown should be shown - qPCR or western.

Answer: *Western blot experiments have not been performed due a lack of relevant and commercially available antibodies suitable for recognition of specific C. elegans proteins.*

Quantitative PCRs were performed on all RNAi constructs, which we obtained from the Ahringer and Vidal RNAi libraries, however this proved to be technically challenging since these RNAi constructs are frequently full length. Therefore, our qPCR analysis led to the detection of the RNAi construct taken up by the nematode rather than the detection of the destruction of the targeted RNA sequence. Nevertheless, considering the data presented in this manuscript and the observed effects on short term brood size and/or long term lifespan experiments, at least partial knockdown of the targeted proteins can be expected. Moreover, as the abovementioned library constructs are widely used sources for knockdown, many of the constructs used in this manuscript haven been used successfully elsewhere.

Reviewer #2 - 14: Supp. Fig 6 - does 'other' mean unchanged in both sets? Or is there only a 5% overlap with the Murphy study- not in high concordance as suggested?

Answer: *In this graph the category "Other" indicates the genes/proteins which were differentially expressed (DE) on the mRNA (Murphy et al, 2003) or protein level (Dong et al, 2007) respectively, but which were not found DE on the protein level in here presented quantitative proteomics data set.*

The overlap found between the presented dataset in this paper and the mRNA expression profiling data (Murphy et al, 2003) is indeed only 5%. However, this overlap is still statistically significant as determined by Hypergeometric statistical testing using all genes in the genome as a background set (Suppl. Fig. S6A). Of note, the study presented by Murphy et al (2003) profiles 19.000 genes, compared to the 455 proteins (Fig. 1b) that were quantified in our dataset which is far from quantification of all proteins expressed in C. elegans.

Importantly, the genes which did overlap between the studies displayed a good concordance in their direction and degree of regulation (Suppl. Fig S6B). That is, genes going down in their mRNA expression generally also went down in our proteomics experiments and vice versa. Through Gene Set Enrichment Analysis (GSEA) we learned that this overlap indeed was statistically significant ($p < 0.001$). Such concordance is not routinely found in these sorts of comparative 'omics' approaches and therefore emphasizes the quality of our resource.

In the revised manuscript "other" is now specified in the legend of Suppl. Fig. S6A (page 2 of Suppl. Information file) where we additionally emphasize the comparison is based on 19.000 genes (Murphy et al, 2003) versus 455 proteins in the here presented study. Moreover, the manuscript has been rephrased: "good concordance" has been replaced by "significant concordance" (page 7, 1st paragraph).

Reviewer #2 - 15: Does the addition of FUDR or Ampicillin affect lifespan? Either by regulating DNA replication (endoreduplication or egg production?) or gut fauna? Is there an effect on lifespan of using non-dividing OP50?

Answer: *Lifespan assays were performed according to a standard protocol (Hansen et al, 2005; Hansen et al, 2007). This protocol includes FudR for chemical sterilization to avoid contamination of the experiment with progeny. The use of FudR for longevity analysis does not affect adult life span (Mitchell et al, 1979; Sutphin & Kaerberlein, 2009), and is hence widely used in lifespan assays.*

Ampicillin is a resistance marker to prevent foreign bacterial contamination and select for the presence of RNAi vector containing bacteria (HT115), critical for targeted knockdown of gene transcription. In the presence of RNAi, bacteria expressing these constructs are resistant to ampicillin due to a copy of the ampicillin resistance gene on both vector control (L4440) and RNAi constructs. The average median lifespan of N2 nematodes on this HT115 strain expressing L4440 is 13.1 days or 13.7 days when switched to this food source in L4 or YA stage respectively (Suppl. Table IVB). These values do not differ from the median lifespan observed for N2 nematodes normal food source (dividing OP50) and performed without ampicillin added to the NGM plates (Suppl. Table IVB) which demonstrates the absence of an effect of Ampicillin on nematode longevity.

Finally, to the best of our knowledge nematodes do not have a gut fauna but can accumulate ingested bacteria upon age (Portal-Celhay et al, 2012). Many parameters affect C. elegans lifespan including temperature, bacterial food strain, genetic background, dividing or non-dividing food. The applied protocol prescribes dividing bacteria as food source, a common protocol for feeding C. elegans in lifespan assays and of eminent importance when RNAi constructs need to be expressed. Importantly, since identical conditions are used for all our (mutant) comparisons and RNAi experiments, we are convinced that our results are not caused by experimental artifacts.

REFERENCES

- Arantes-Oliveira N, Apfeld J, Dillin A, Kenyon C (2002) Regulation of life-span by germline stem cells in *Caenorhabditis elegans*. *Science* **295**: 502-505
- Dillin A, Crawford DK, Kenyon C (2002) Timing requirements for insulin/IGF-1 signaling in *C. elegans*. *Science* **298**: 830-834
- Dong MQ, Venable JD, Au N, Xu T, Park SK, Cociorva D, Johnson JR, Dillin A, Yates JR, 3rd (2007) Quantitative mass spectrometry identifies insulin signaling targets in *C. elegans*. *Science* **317**: 660-663
- Gems D, Sutton AJ, Sundermeyer ML, Albert PS, King KV, Edgley ML, Larsen PL, Riddle DL (1998) Two pleiotropic classes of *daf-2* mutation affect larval arrest, adult behavior, reproduction and longevity in *Caenorhabditis elegans*. *Genetics* **150**: 129-155
- Ghazi A, Henis-Korenblit S, Kenyon C (2007) Regulation of *Caenorhabditis elegans* lifespan by a proteasomal E3 ligase complex. *Proceedings of the National Academy of Sciences of the United States of America* **104**: 5947-5952
- Hansen M, Chandra A, Mitic LL, Onken B, Driscoll M, Kenyon C (2008) A role for autophagy in the extension of lifespan by dietary restriction in *C. elegans*. *PLoS genetics* **4**: e24
- Hansen M, Hsu AL, Dillin A, Kenyon C (2005) New genes tied to endocrine, metabolic, and dietary regulation of lifespan from a *Caenorhabditis elegans* genomic RNAi screen. *PLoS genetics* **1**: 119-128
- Hansen M, Taubert S, Crawford D, Libina N, Lee SJ, Kenyon C (2007) Lifespan extension by conditions that inhibit translation in *Caenorhabditis elegans*. *Aging cell* **6**: 95-110
- Henis-Korenblit S, Zhang P, Hansen M, McCormick M, Lee SJ, Cary M, Kenyon C (2010) Insulin/IGF-1 signaling mutants reprogram ER stress response regulators to promote longevity. *Proceedings of the National Academy of Sciences of the United States of America* **107**: 9730-9735
- Hirsh D, Oppenheim D, Klass M (1976) Development of the reproductive system of *Caenorhabditis elegans*. *Developmental biology* **49**: 200-219
- Hsin H, Kenyon C (1999) Signals from the reproductive system regulate the lifespan of *C. elegans*. *Nature* **399**: 362-366
- McCulloch D, Gems D (2003) Body size, insulin/IGF signaling and aging in the nematode *Caenorhabditis elegans*. *Experimental gerontology* **38**: 129-136
- Melendez A, Talloczy Z, Seaman M, Eskelinen EL, Hall DH, Levine B (2003) Autophagy genes are essential for dauer development and life-span extension in *C. elegans*. *Science* **301**: 1387-1391
- Mitchell DH, Stiles JW, Santelli J, Sanadi DR (1979) Synchronous growth and aging of *Caenorhabditis elegans* in the presence of fluorodeoxyuridine. *Journal of gerontology* **34**: 28-36

Murphy CT, McCarroll SA, Bargmann CI, Fraser A, Kamath RS, Ahringer J, Li H, Kenyon C (2003) Genes that act downstream of DAF-16 to influence the lifespan of *Caenorhabditis elegans*. *Nature* **424**: 277-283

Portal-Celhay C, Bradley ER, Blaser MJ (2012) Control of intestinal bacterial proliferation in regulation of lifespan in *Caenorhabditis elegans*. *BMC microbiology* **12**: 49

Sutphin GL, Kaerberlein M (2009) Measuring *Caenorhabditis elegans* life span on solid media. *Journal of visualized experiments : JoVE*

Thank you again for submitting your work to Molecular Systems Biology. We have now heard back from the two referees who accepted to evaluate the revised study. As you will see from the reports below, the reviewers acknowledge the efforts made during the revision of this study. Nonetheless, the referees still raise substantial concerns on the major conclusions of your work. In particular, the major issue raised by both reviewers in the previous round remains unfortunately inconclusively addressed. Thus, reviewer #1 thinks that there is a "very strong possibility that the effects on translation-associated proteins occur mainly in the germline" and, similarly, reviewer 2 feels that this "major difficulty [...] remains largely unresolved". The reviewers are thus still not convinced that the study provides the "airtight arguments" requested in the first round. As such, I am afraid that the level of support provided by the reviewers remains too limited for publication in Molecular Systems Biology.

Under these circumstances, I see no other choice than to return the manuscript with the message that we cannot offer to publish it. In any case, thank you for the opportunity to examine your work. I hope that the points raised in the reports will prove useful to you and that you will not be discouraged from submitting future work to Molecular Systems Biology.

Reviewer #1 (Remarks to the Author):

This paper has been improved substantially by addition of new data and enhanced clarity in presentation of existing data. The proteomics results themselves are solid, of wide interest, and should be of high priority for publication. The followup experiments are for the most part also of significant interest, although many of them are still overinterpreted in a surprisingly uncritical manner. For the most part the authors have responded satisfactorily to my concerns, but serious problems in interpretation of the data and literature remain that preclude publication of the paper in its current version. It should be possible to address these issues, which mainly concern the issue of overinterpretation and the major caveat that the data still do not eliminate the very strong possibility that the effects on translation-associated proteins occur mainly in the germline.

Concerns include:

It should be stated in the text that proteomic analyses were performed at 25 degrees, since this could have influenced the results. In the cited Gems paper, *daf-2* Class 2 traits (body size, brood size, movement) that are seen in *e1370* are all much more prominent at 25 degrees, as is the tendency to enter dauer. The effects of *daf-2* mutations are therefore temperature-dependent, making it important that the reader consider temperature when interpreting the reported findings. This does not make these findings invalid at all, but the informed reader should not have to dig to find out the conditions used. Similarly, the stage of animals analyzed should also be stated prominently, since the results really provide a temporal snapshot.

The most serious problem is that the authors persist in largely ignoring the fact that they identified a number of germline-specific proteins in their analysis, which creates a real cause for worry with respect to their interpretation. PGL-1 is germline specific, CGH-1 is expressed mostly in the germline, and I suspect that this would be true for PGY-4 and even PAB-1. As noted, these proteins even form complexes in the germline (which are much more prominent than any somatic counterparts). This strongly suggests that effects on translation factors may also occur primarily in the germline, especially since the analysis was performed at 25 degrees (see above), a condition in which *e1370* shows reproductive abnormalities. It is very helpful that the authors show that translation is affected at lower temperature as well, but the authors really must show more caution in interpretation. This still does not indicate that "reduced protein translation can induce longevity independently of reproduction" (pg. 12). The lifespan data with *pgl-1*, *cgh-1*, *car-1*, etc does not add support for their model, to the contrary, it suggests that interference with germline mRNA metabolism/translation increases lifespan (they even show reduced brood sizes for some of these!). True, this is *daf-16*-dependent, but so is the germline pathway identified by Kenyon. It is therefore

particularly misleading to speculate about the involvement of RNA bodies in the discussion, when most of the proteins involved are expressed mainly in the germline. The single piece of data that supports their model (but wouldn't prove it) is the analysis of aars-2 RNAi, which decreases lifespan to a very minimal extent in two experiments that differ from each other substantially in their control values. These data are not very convincing but might be right, but would still not be conclusive enough to hang the central model of the paper on. It was a step in the right direction to change the title and soften the model somewhat, but it still diminishes the value of the work to gloss over and selectively interpret the prominent evidence that something is likely happening in the germline here. The paper is still publishable, but the work will be a lot more valuable to the community if this is presented in a more balanced manner.

Presentation of the possible link between the insulin/IGF-1 and TOR pathways is garbled and misleading - in contrast to mammals, in *C. elegans* there is no direct evidence that insulin/IGF signaling controls the TOR pathway. Although this seems likely based on findings in other species, *C. elegans* lacks known TSC proteins, which would provide this link. Here as in other places, a failure to extend lifespan in an additive manner is overinterpreted. More caution is warranted.

Pg. 5 - comparison to the daf-2;daf-16 double mutant does NOT exclude daf-2-mediated longevity effects that are independent of longevity assurance, because dauer and all of the daf-2 Class 2 traits (see Gems paper) are daf-16-dependent.

Pg. 3 - daf-2 longevity is not suppressed completely by skn-1/Nrf, mutation of which barely affects daf-2(e1370) at all according to published data.

Pg. 11 - decreased longevity of proteasomal subunits (??).

Pg. 14 - what is the evidence that let-363 is "thought to mediate *C. elegans* Insulin/IGF-1 longevity"?

Pg. 16 - "DAF-2 is essential during dauer formation". DAF-2 activity actually inhibits dauer formation. I do not understand the next sentence that begins with "Dauer larvae".

Reviewer #2 (Remarks to the Author):

In general the authors have made a genuine effort to respond to the reviewers' comments with new data, re-writing of parts of the MS and reasoned argument, and have thus dealt with many of the original comments adequately. However, the major difficulty pointed out by both reviewers remains largely unresolved, which is where the protein metabolic signal is coming from. The evidence put forward by the authors does not clinch this point, since it is all indirect. The only way to sort this out would be to do the full set of experiments with the germ line absent or inhibited in proliferation, to pin the signals on the somatic cells. As far as I can see, the important and relevant point of Reviewer 1 about the Ruvkun lab findings remains unaddressed.

Detailed point to point response:

Belonging to Stout *et al.*: Insulin mediated longevity is marked by reduced protein metabolism
MSB 12-3940 / MSB- 13 4491

Reviewer #1 (Remarks to the Author):

This paper has been improved substantially by addition of new data and enhanced clarity in presentation of existing data. The proteomics results themselves are solid, of wide interest, and should be of high priority for publication. The followup experiments are for the most part also of significant interest, although many of them are still overinterpreted in a surprisingly uncritical manner. For the most part the authors have responded satisfactorily to my concerns, but serious problems in interpretation of the data and literature remain that preclude publication of the paper in its current version. It should be possible to address these issues, which mainly concern the issue of overinterpretation and the major caveat that the data still do not eliminate the very strong possibility that the effects on translation-associated proteins occur mainly in the germline.

Concerns include:

It should be stated in the text that proteomic analyses were performed at 25 degrees, since this could have influenced the results. In the cited Gems paper, *daf-2* Class 2 traits (body size, brood size, movement) that are seen in *e1370* are all much more prominent at 25 degrees, as is the tendency to enter dauer. The effects of *daf-2* mutations are therefore temperature-dependent, making it important that the reader consider temperature when interpreting the reported findings. This does not make these findings invalid at all, but the informed reader should not have to dig to find out the conditions used. Similarly, the stage of animals analyzed should also be stated prominently, since the results really provide a temporal snapshot.

Indeed, temperature and stage are important when interpreting our data. We have now added this information (temperature and animal stage) to the main text where we introduce the proteomics experiment in the results section.

The most serious problem is that the authors persist in largely ignoring the fact that they identified a number of germline-specific proteins in their analysis, which creates a real cause for worry with respect to their interpretation. PGL-1 is germline specific, CGH-1 is expressed mostly in the germline, and I suspect that this would be true for PGY-4 and even PAB-1. As noted, these proteins even form complexes in the germline (which are much more prominent than any somatic counterparts). This strongly suggests that effects on translation factors may also occur primarily in the germline, especially since the analysis was performed at 25 degrees (see above), a condition in which *e1370* shows reproductive abnormalities. It is very helpful that the authors show that translation is affected at lower temperature as well, but the authors really must show more caution in interpretation. This still does not indicate that "reduced protein translation can induce longevity independently of reproduction" (pg. 12). The lifespan data with *pgl-1*, *cgh-1*, *car-1*, etc does not add support for their model, to the contrary, it suggests that interference with germline mRNA metabolism/translation increases lifespan (they even show reduced brood sizes for some of these!). True, this is *daf-16*-dependent, but so is the germline pathway identified by Kenyon. It is therefore particularly misleading to speculate about the involvement of RNA bodies in the discussion, when most of the proteins involved are expressed mainly in the germline. The single piece of data that supports their model (but wouldn't prove it) is the analysis of *aars-2* RNAi, which decreases lifespan to a very minimal extent in two experiments that differ from each other substantially in their control values. These data are not very

convincing but might be right, but would still not be conclusive enough to hang the central model of the paper on. It was a step in the right direction to change the title and soften the model somewhat, but it still diminishes the value of the work to gloss over and selectively interpret the prominent evidence that something is likely happening in the germline here. The paper is still publishable, but the work will be a lot more valuable to the community if this is presented in a more balanced manner.

We have performed a novel experiment in which we ablated the daf-2 germline and observed that the prominent reduction in protein translation in the daf-2 mutant is independent of the germline (new Fig. 4). Moreover, we ablated the germline of wild-type nematodes and this does not decrease global protein translation. Thus, these experiments conclusively demonstrate that the reduced protein translation phenotype stems from the somatic tissue and furthermore suggest that longevity due to germline ablation is likely mediated independently of a global reduction in protein translation. However, we indeed observed a decrease in proteins in the daf-2 mutant that are germline enriched/specific proteins like PGL-1, CGH-1 etc, in agreement with the observed reduced fecundity phenotype of the daf-2 mutant. Moreover, knockdown of these proteins extends longevity and these findings indeed suggest that at least part of the strong daf-2 longevity phenotype is mediated by decreased germline activity, although the reduced global protein translation phenotype is mediated predominantly by somatic tissue. We have now included a full novel discussion section to discuss the germ-line contribution.

Other changes in the manuscript include:

-The discussion on RNA bodies has been eliminated as it was predominantly speculation.

-The claim at p12 "reduced protein translation can induce longevity independently of reproduction" has been changed into "reduced protein translation can induce longevity".

Presentation of the possible link between the insulin/IGF-1 and TOR pathways is garbled and misleading - in contrast to mammals, in *C. elegans* there is no direct evidence that insulin/IGF signaling controls the TOR pathway. Although this seems likely based on findings in other species, *C. elegans* lacks known TSC proteins, which would provide this link. Here as in other places, a failure to extend lifespan in an additive manner is overinterpreted. More caution is warranted.

We apologize for this and have completely rewritten the TOR part of the discussion. We have also eliminated the speculation on RNA-bodies.

Pg. 5 - comparison to the *daf-2;daf-16* double mutant does NOT exclude *daf-2*-mediated longevity effects that are independent of longevity assurance, because dauer and all of the *daf-2* Class 2 traits (see Gems paper) are *daf-16*-dependent.

We have removed this sentence.

Pg. 3 - *daf-2* longevity is not suppressed completely by *skn-1/Nrf*, mutation of which barely affects *daf-2(e1370)* at all according to published data.

We have changed this into "partially suppressed".

Pg. 11 - decreased longevity of proteasomal subunits (??).

Indeed, this is an unfortunate sentence and this has been changed.

Pg. 14 - what is the evidence that let-363 is "thought to mediate C. elegans Insulin/IGF-1 longevity"?

We apologize if this suggestion was provoked by our choice of words. We have changed this and as mentioned above, we have now completely rewritten the TOR discussion.

Pg. 16 - "DAF-2 is essential during dauer formation". DAF-2 activity actually inhibits dauer formation. I do not understand the next sentence that begins with "Dauer larvae".

The part of the dauer formation has been removed from the discussion as it was predominantly speculation.

Reviewer #2 (Remarks to the Author):

In general the authors have made a genuine effort to respond to the reviewers' comments with new data, re-writing of parts of the MS and reasoned argument, and have thus dealt with many of the original comments adequately. However, the major difficulty pointed out by both reviewers remains largely unresolved, which is where the protein metabolic signal is coming from. The evidence put forward by the authors does not clinch this point, since it is all indirect. The only way to sort this out would be to do the full set of experiments with the germ line absent or inhibited in proliferation, to pin the signals on the somatic cells. As far as I can see, the important and relevant point of Reviewer 1 about the Ruvkun lab findings remains unaddressed.

We thank the reviewer for acknowledging our effort to respond to the reviewers' concerns. The reviewer however isn't fully convinced yet and raises two points:

1) the relevant point of reviewer 1 about the Ruvkun lab findings hasn't been addressed.

The Ruvkun lab published expression of germline genes in abnormally high levels in somatic tissue (Curran, Nature 2009) and the question of reviewer 1 is whether we did not detect an increase in PGL-1 because the observed reductions of PGL-1 occurred predominantly in the germline. It is indeed possible that the observed decrease in PGL-1 in our proteomics dataset is the result of the balance between a PGL-1 downregulation in the germline versus a PGL-1 upregulation in the soma, resulting in a net decrease in PGL-1 expression. Importantly however, we agree with both reviewers that the daf-2 mutant shows a reproduction phenotype, which is at least in part responsible for its longevity phenotype. We have discussed this now in a new discussion chapter.

Although downregulation of germline enriched/specific proteins were observed, we now show with a novel experiment using germline ablation, that the daf-2 global downregulation of protein translation is not mediated by the germline (see below).

2) Where is the protein metabolic signal coming from? The reviewer points that we only provided indirect evidence and asks for a full set of experiments to pin the effect on the somatic cells.

We have performed a novel experiment in which we ablated the daf-2 germline and observed that the prominent reduction in protein translation in the daf-2 mutant is independent of the germline (new Fig. 4). Moreover, we ablated the germline of wild-type nematodes and this does not decrease global protein translation. Thus, these experiments conclusively demonstrate that the reduced protein translation phenotype stems from the somatic tissue and furthermore suggest that longevity due to germline ablation is likely mediated independently of a global reduction in protein translation. However, we indeed observed a decrease in proteins in the daf-2 mutant that are germline enriched/specific proteins like PGL-1, CGH-1 etc, in agreement with the observed reduced fecundity phenotype of the daf-2 mutant. Moreover, knockdown of these proteins extends longevity and these findings indeed suggest that at least part of the strong daf-2 longevity phenotype is mediated by decreased germline activity, although the reduced global protein translation phenotype is mediated predominantly by somatic tissue. We have now included a full novel discussion section to discuss the germ-line contribution.

3rd Editorial Decision

21 May 2013

Thank you again for submitting your work to Molecular Systems Biology. We have now heard back from the referee who accepted to evaluate the study. As you will see, this referee is supportive but requests a specific amendment to the text (see report below). We would also ask you to include in Materials and Methods a section on the GLP-1 siRNA experiment and indicate explicitly how germline ablation was verified upon siRNA against GLP-1 siRNA to make sure the procedure worked.

Please resubmit your revised manuscript online as soon as possible, with a covering letter listing amendments and responses to each point raised by the referees.

Reviewer #1 (Remarks to the Author):

With the new experiments the authors have addressed my major concerns, but I would like one issue to be clarified in the text before publication. At the bottom of pg 14 they group results obtained at 15 and 20 degrees together in a misleading way. Reproductive phenotypes are not observed at 15 but do occur at 20. At the same time, they see no translation reduction at 15 and the reduction at 20 is not as striking as was seen at 25. The translation reduction therefore tracks with the severity of the reproductive phenotype. This does not affect the validity or suitability of their work for publication in MSB, in light of their other new results, but they should not discuss these particular data in this way.

3rd Revision - authors' response

24 May 2013

Manuscript: MSB-12-3940RR

Detailed point to point response:

Reviewer #1 (Remarks to the Author):

With the new experiments the authors have addressed my major concerns, but I would like one issue to be clarified in the text before publication. At the bottom of pg 14 they group results obtained at 15 and 20 degrees together in a misleading way. Reproductive phenotypes are not observed at 15 but do occur at 20. At the same time, they see no translation reduction at 15 and the reduction at 20 is not as striking as was seen at 25. The translation reduction therefore tracks with the severity of the reproductive phenotype. This does not affect the validity or suitability of their work for publication in MSB, in light of their other new results, but they should not discuss these particular data in this way.

*Although considered not significant ($p < 0.05$) by Gems et al, Genetics 1998, recalculation of their data shows a small difference that reaches statistical significance at $p = 0.05$, student t-test. Also Kenyon et al, Nature 1993 reported a small decrease in broodsize for the *daf-2(e1370)* at 20°C. We have therefore rephrased the text at the bottom of p14 and follow the reviewers' suggestion to conclude that the translation reduction therefore tracks with the severity of the reproductive phenotype:*

*"Next, polyribosome analysis was performed in the *daf-2(e1370)* mutant propagated at 15°C and 20°C when a longevity phenotype is observed either in the absence (15°C) or with a small (20°C) reproductive effect (Gems et al, 1998); (Kenyon et al, 1993). A reduction in active protein translation was found (**Suppl. Fig. S10**), although the magnitude of the reduction tracks with the severity of the reproductive phenotype."*

Because of this amendment we have removed this argument from the discussion section "Reproduction and Insulin/IGF-1 mediated longevity" at page 18.

We would also ask you to include in Materials and Methods a section on the GLP-1 siRNA experiment and indicate explicitly how germline ablation was verified upon siRNA against GLP-1 siRNA to make sure the procedure worked. We have added the following paragraph to the Materials and Methods section (page 24, third paragraph from the top):

*Germline ablation. Synchronized L1 nematodes were plated on vector control or *glp-1* RNAi-containing HT115 bacteria plates, incubated at 15°C until the worms reached L4 stage, then temperature shifted to 25°C for 16-20 hours, harvested and subsequently snap frozen for polysome profiling. Ablation of the germline of adult nematodes was confirmed by Nomarsky microscopy before the nematodes were harvested.*